# Metal Organic Framework Based Polymer Mixed Matrix Membranes: Review on Applications in Water Purification

**DOI:** 10.3390/membranes9070088

**Published:** 2019-07-19

**Authors:** Asmaa Elrasheedy, Norhan Nady, Mohamed Bassyouni, Ahmed El-Shazly

**Affiliations:** 1Chemical and Petrochemicals Engineering Department, Egypt-Japan University of Science and Technology (E-JUST), Alexandria 21934, Egypt; 2Department of Chemical Engineering, Faculty of Engineering, Port Said University, Port Said 42526, Egypt; 3Polymeric Materials Research Department, City of Scientific Research and technological Applications (SRTA-city), Borg El-Arab City, Alexandria 21934, Egypt; 4Materials Science Program, Zewail University of Science and Technology, City of Science and Technology, October Gardens, 6th of October, Giza 12578, Egypt; 5Chemical Engineering Department, Faculty of Engineering, Alexandria University, Alexandria 21544, Egypt

**Keywords:** metal-organic framework, water purification, desalination, composite membranes

## Abstract

Polymeric membranes have been widely employed for water purification applications. However, the trade-off issue between the selectivity and permeability has limited its use in various applications. Mixed matrix membranes (MMMs) were introduced to overcome this limitation and to enhance the properties and performance of polymeric membranes by incorporation of fillers such as silica and zeolites. Metal-organic frameworks (MOFs) are a new class of hybrid inorganic–organic materials that are introduced as novel fillers for incorporation in polymeric matrix to form composite membranes for different applications especially water desalination. A major advantage of MOFs over other inorganic fillers is the possibility of preparing different structures with different pore sizes and functionalities, which are designed especially for a targeted application. Different MMMs fabrication techniques have also been investigated to fabricate MMMs with pronounced properties for a specific application. Synthesis techniques include blending, layer-by-layer (LBL), gelatin-assisted seed growth and in situ growth that proved to give the most homogenous dispersion of MOFs within the organic matrix. It was found that the ideal filler loading of MOFs in different polymeric matrices is 10%, increasing the filler loading beyond this value led to formation of aggregates that significantly decreased the MOFs-MMMs performance. Despite the many merits of MOFs-MMMs, the main challenge facing the upscaling and wide commercial application of MOFs-MMMs is the difficult synthesis conditions of the MOFs itself and the stability and sustainability of MOFs-MMMs performance. Investigation of new MOFs and MOFs-MMMs synthesis techniques should be carried out for further industrial applications. Among these new synthesis methods, green MOFs synthesis has been highlighted as low cost, renewable, environmentally friendly and recyclable starting materials for MOFs-MMMs. This paper will focus on the investigation of the effect of different recently introduced MOFs on the performance of MOFs-MMMs in water purification applications.

## 1. Introduction

One of the major solutions to the water scarcity problem is desalination and wastewater treatment [1,2,3]. However, wastewater treatment includes many steps to produce usable water. Thermal-based and membrane-based desalination processes are competitive, but membrane-based processes have proved to be the most cost-efficient desalination processes [4]. Water purification using membrane technology becomes a vital solution of several processes including food, drugs, dairy, cosmetics and chemical industries [5,6]. Since 1960s, when Loeb and Sourirajan introduced a successful skin cellulose acetate membrane [6] many organic [7,8] and inorganic [9] membranes have been investigated for different applications, especially for desalination. After producing the first thin film composite (TFC) reverse osmosis (RO) membrane in 1980s, many studies were focused on enhancing the membrane properties and performance by fabricating mixed matrix membranes (MMM) by adding certain fillers [10] such as zeolites [11,12,13,14,15], silica [16,17,18,19,20,21,22] and titanium oxide [23,24,25,26].

Recently, metal-organic frameworks (MOFs) have been introduced as novel membranes’ fillers [27], pure MOFs thin films [27,28], or as thin films on organic [29,30] and inorganic [31] substrates for different application such as gas separation [32,33,34,35] and desalination [36,37,38,39,40,41,42]. MOFs are the combination of organic ligands and metal-containing nodes that are formed by directional coordination bonds. The synergistic effect of the two components of MOFs, which are the organic ligand and the inorganic metal ion or cluster, provide exceptional properties. These unique properties include adjustable pore size, internal surface areas, enormous porosity [43], regularity, rigidity/flexibility, and versatile structures [44]. These exceptional and unique properties attracted the interest for MOFs utilization in applications such as high capacity adsorbents [45,46,47,48,49], organic contaminants removal [50,51], environmental protection [52], heavy metals removal [53,54,55], H_2_ storage [56,57,58,59,60] and purification [61], CO_2_ adsorption [62,63] and capture [64,65,66,67,68,69,70], drug delivery [71,72,73,74,75,76,77,78], gas/liquid separation [79,80,81,82], membrane fuel cell [83], organic solvent nanofiltration [84,85,86,87] and storage media [88,89,90,91].

The presence of the organic ligands in the structure of MOFs offers many advantages over other inorganic nanostructured fillers used in polymeric matrices. Since the organic linkers present in MOFs have a higher affinity for polymeric chains, the control of the interactions and the gaps at the polymer-MOFs interface can be easily controlled. Post-synthetic treatment can also be used to accommodate the produced MOFs for a particular application by altering their chemical design, pore size and shape [92]. Incorporation of MOFs in polymeric membranes for liquid separation processes has not been reported in literature as much as other separation processes, therefore this paper will focus on MOFs composite polymeric membranes used in water treatment and desalination.

## 2. Timeline and Synthesis Routes

Metal-organic frameworks (MOFs) are made up of uniform infinite repeating units of organic linkers and inorganic nodes. MOFs were also recognized as crystalline materials which have very high porosity (up to 90% free volume) and huge internal surface areas (over 6000 m^2^/g) [43,44]. Recently, MOFs are being referred to as materials formed by strong covalent bonding between secondary building units that are polyatomic metal-containing clusters and rigid organic moieties [44].

MOFs synthesis field was known to emerge from the co-ordination and solid-state/zeolite chemistry field. In 1989 and 1990, Hoskins and Robson envisioned the MOFs future which was the possibility of manufacturing of a wide range of crystalline, microporous, stable solids, possibly using structure-directing agents, with ion-exchange, gas sorption, or catalytic properties, where functional groups can be added by post-synthesis modification. This vision was later proved by scientists all over the world. In 1997, Kitagawa et al. introduced a 3D MOF which showed gas sorption properties at room temperature [93]. The milestone MOF-5 and HKUST-1 were reported in 1999 and up until now, these two types are among the most widely studied MOFs [94]. Shortly thereafter, the highly stable MIL-101 emerged [44] while zeolite imidazole frameworks (ZIFs) were first introduced in 2002 among the imidazolate-based compounds [94]. A large number of MOFs have been developed since then. However, researchers have been focused on the modification and further application of already developed highly stable MOFs such as ZIFs, HKUST-1, UiO-66, Cr-Mil-101, Cr-MIL-100, MIL-125, etc.

MOFs synthesis is usually carried out by one-pot self-assembly reactions between solutions of metal salts and the organic linkers, at a temperature ranging from room temperature to 250 °C. At low temperatures, the single crystal growth is promoted due to the low evaporation rate of solvents. Whereas, at higher temperature and pressure conditions (i.e., solvothermal technique), production of single crystalline MOFs is observed at lower reaction time. MOFs obtained via solvothermal technique are more complicated than those MOFs obtained at room temperature; therefore, to obtain high quality MOFs, controlling the reaction rate by adjusting different reaction parameters is essential. At the same time, microwave-assisted MOF growth proved to deliver superior outcomes when compared to other synthetic methods especially for quick synthesis. Ionothermal MOF synthesis (where an ionic liquid that acts as both solvent and template) was also reported. Stepwise approaches for MOFs synthesis proved to deliver metal–organic polyhedrals (MOPs) with a higher degree of controllability. Nanoscale MOFs have also attracted a lot of attention recently due to their outstanding characteristics [44]. The design-ability and adjustability of MOFs are considerably higher than other porous fillers, this may be attributed to many parameters as follows: the controllable (mild) synthetic conditions of MOFs, the ability to modify the organic ligands, the fixed coordination geometries, the ability of predesigning the organic linkers and changing the synthesis conditions without altering their connectivity or topology, and post-synthetic techniques can be used to modify the metal-clusters or the organic ligands [44].

A new approach in MOFs synthesis has emerged during the last decade, which is the “green” synthesis of MOFs. The essential reason that triggered this approach is that the conventional MOFs synthesis methods cannot be applied on large commercial scale due to unsustainable and unsuitable synthesis conditions. In other words, the “green” approach emerged essentially from the need of cheap, renewable, and recyclable starting materials that avoided waste while saving energy to be applied industrially. The term “green” in MOFs synthesis can be described in the following simple points: (1) Avoid excessive use of solvents during synthesis or fabrication to minimize waste considerably. (2) Highly selective and high through output synthesis techniques should be adopted to maximize production and avoid any by-products. (3) Nontoxic and harmless reactants to human health and the environment and simple precursors should be used in synthesis techniques whenever applicable and practical. (4) The synthesis routes should preserve the targeted properties of the products. (5) Synthesis techniques employing low energy requirements should be applied where MOFs would be fabricated at atmospheric pressure and ambient temperature. (6) Renewable sources for the organic linkers (such as starch and cellulose) and solvents should be used whenever technically and economically viable. (7) Products should not be environmentally persistent and to be degraded under working conditions. (8) Development of new controllable synthesis techniques comprising shorter reaction times to avoid formation of side products as much as possible. (9) During scaling up MOFs synthesis, development of new chemical methods should be chosen to avoid chemical accidents such as explosions and releases. For example, green techniques were used to synthesize green ZIF-8. Alternative less hazardous solvents such as methanol were used instead of N,N-dimethylformamide (DMF). A major advantage of this approach is only water is formed as a byproduct. However, the porosity of the formed ZIF-8 was much lower than expected. Mechanical techniques by using ball milling were also reported for the synthesis of ZIF-8. In this case, the issue was that reaction took a very long time to produce ZIF-8 with the highest porosity and that a core of zinc oxide was found within the ZIF-8 shells, hence the reaction was not fully achieved. However, all the reported procedures utilizing lower amounts of solvents or no solvent at all requires the removal of residual linkers by solvent treatment after synthesis [95].

Green water-soluble MOFs have also been applied as green templates for the formation of macropores to produce porous matrix membranes (PMM) with enhanced porosity for water treatment [96,97]. 

MOFs-MMM synthesis routes can be categorized into four general routes; the blending technique, in situ growth, layer-by-layer (LBL) and gelatin-assisted seed growth. Table 1 summarizes the procedures of the four general synthesis routes. The blending technique is divided into three different dope routes. The main difference between these dope preparation routes is the method of doping the filler into the polymer matrix. Mixed matrix dope preparation routes used in the blending technique are illustrated in Figure 1.

Table 2 presents the different kinds of fillers that were focused on this review. Membrane performance, filler loading, rejection, application and selection criteria are also provided for a more comprehensive overview for the points that will be covered throughout the next section.

## 3. MOFs Applications in Water Purification Membranes

There are many types of MOFs that have been investigated and used as fillers for organic membranes to synthesize MMMs to be used in water purification such as zeolite imidazolate frameworks (ZIFs), materials of Institute Lavoisier (MIL), University of Oslo (UiO-66), and green MOFs (F300, A100 and C300) [110,111]. In the following section, we will focus on the most common MOFs which are ZIFs, UiO-66 and green MOFs.

### 3.1. Zeolite Imidazolate Frameworks (ZIFs)

Zeolite imidazolate frameworks (ZIFs) are crystalline porous and combine the desirable properties of both zeolites and MOFs. These properties include high surface area, microporosity, thermal and chemical stability. The nanosized pores in the structure of ZIFs are made up by linking a metal ion such as Zn, Cu and Co by either a functionalized imidazolate (Im) or ditopic Im links through nitrogen atoms to form four, six, eight and twelve rings of ZnN_4_, CuN_4_ and CoN_4_ tetrahedral clusters. The structure and the bond angles of both zeolites and ZIFs are very much alike, zeolites contain T–O–T bridges where T = Si, Al or P while ZIFs contains M–Im–M bridges where M = Zn, Cu, or Co; the bond angles in both structures are 145° [112].

#### ZIF-8

ZIF-8 is the most famous and widely used of this family due to its stable tetrahedral MN_4_ structure and its hydrophobic pores, which make it chemically and thermally stable and resistant to water and organic solvents [35]. The narrow window size (3.4 Å) of ZIF-8 and high specific surface area (~1400 m^2^/g) also made it a very good candidate for polyamide (PA) thin film nanocomposite (TFN) membranes [29]. A schematic representation of ZIF-8 structure is shown in Figure 2.

For the previous mentioned reasons, Lee et al. studied the effect of ZIF-8 particle size (60, 150 and 250 nm) on the performance of polyamide (PA) thin-film nanocomposite reverse osmosis (RO) membranes. ZIF-8 nanoparticles were first dispersed in aqueous solution of m-phenylene diamine (MPD) by concentration of 0.2% *w/v*. The ZIF-8 nanoparticles were found to disperse well and form a stable solution in water without any aggregations compared to other organic solvents such as n-hexane or n-decane. This may be due to the strong electrostatic interactions between the positive charges formed on the ZIF-8 in aqueous solution and water molecules. It was found that TFN membranes containing the medium sized ZIF-8 (ZIF-8(M)) particles showed the highest enhancement of water permeance of 3.95 L/m^2^·h·bar with salt rejection of 99.2% compared to the other two sizes. This result may be attributed to the interfacial area between the ZIF-8 and the PA matrix, which was the highest in case of ZIF-8(M). Increasing the concentration of ZIF beyond 0.2% *w/v* (i.e., 0.4% *w/v*) led to severe aggregation of the nanoparticles, accordingly the NaCl rejection was drastically decreased. Another interesting feature about the ZIF-8(M)/PA membranes is that it possessed the lowest activation energy for permeation of molecules through the membrane among other modified membranes and the pristine TFC membrane. The former result may also be attributed to the ZIF-8(M) highest interface area [99].

Kebria et al. synthesized a novel thin film composite (TFC) membrane of PVDF coated with an ultrathin layer of ZIF-8/chitosan and employed it for membrane distillation (MD) for water desalination. The ZIF-8/chitosan layer did not enhance the hydrophobicity of the TFC membrane significantly while the liquid entry pressure of water increased considerably. This may be attributed to the increased surface porosity and decreased mean surface pore size that allowed the permeation of water vapor through the membrane. The permeate water flux also increased up to 350% with 3.5 wt% NaCl feed solution concentration compared to the neat PVDF membrane. The salt rejection was maintained at a reliable value greater than 99.5% using air gap membrane distillation (AGMD) tests at 60 °C. When actual seawater was used as the feed solution, the TFC membrane gave better anti-fouling properties due to the presence of the chitosan layer by a flux reduction of 16% and flux recovery of 90% compared to 41% and 67%, respectively for the neat PVDF membrane [38].

Zhang et al. proposed a strategy for the in situ growth and dispersion of ZIF-8 in polystyrene sulfonate (PSS) polymer namely coordination-driven in situ self-assembly for the synthesis of hybrid ZIF-8/PSS membrane on the surface of a polyacrylonitrile (PAN) support for the removal of methyl blue (MB) dye from water by nanofiltration. In this process, MOFs are produced by covalent coordination between the metal clusters and the organic ligand together with the membrane formation, which result in better dispersion of the produced MOFs in the polymer matrix. The synthesized membrane by the former process exhibits significant enhancement of its compatibility, stability, selectivity, hydrophilicity and adsorption. These changes in membrane properties can be attributed to the coordination bonds generated between the MOFs and the polymer itself. The hydrophilicity of the hybrid membrane was higher than the pure ZIF-8 membranes. It was observed that the particle size of the produced ZIF-8 depends on the initial precursor concentrations; when the concentration of Zn(NO_3_)_2_ was increased from 0.05 to 0.5 mol/L, the particle size decreased from 150 to 50 nm. Also, at the former mentioned concentration range, the membrane surface roughness increased from 0.12 to 0.271 µm and a denser membrane was obtained with a crack free surface. The ZIF-8/PSS hybrid membrane showed elevated flux as well as high rejection values of methyl blue dye (265 L/m^2^·h·MPa and 98.6%, respectively) simultaneously compared to the other purely prepared organic membranes. This observation breaks the trade-off rule of increasing flux on the cost of rejection value and vice versa. On the other hand, the molecular sizes of the dye removed also affected the water flux and the dye rejection, where small molecular sized dyes could permeate through the membrane i.e., lower rejection but the water flux was high. The previous result was confirmed by testing the nanofiltration of methyl orange (MO) dye which has a smaller molecular size than MB and the rejection was 62.4% and 98.6%, respectively [101].

Maroofi et al. prepared (ZIF-8)-Polyvinylpylpyrrolidone (PVP)–Polyethersulfone (PES) hybrid membrane (ZPP) and employed it for the removal of malachite green (MG) dye in a cross-flow system. ZPP composite membranes were fabricated by mixing different concentrations of previously prepared ZIF-8 particles (1%–3%) with PVP/PES and then transformed to membrane films. SEM imaging showed that the ZIF-8 particles were uniformly dispersed within the pores and on the surface of the membrane matrix. This led to increasing the surface roughness and enhancement of the membrane hydrophobicity. ZPP membrane rejection performance was evaluated in a continuous membrane system with flat sheet membrane and MG dye solution of 30 mg/L. Results showed that increasing the ZIF-8 loading in the ZPP membrane increased the dye removal to a maximum of 99.74% at filler loading of 3%. The dye removal rate also was found to increase with increasing the operating time. This increase in performance indicates that the dye removal mechanism is by adsorption of the positively charged cationic dye on the negatively charged adsorption sites [102].

Mao et al. prepared mixed matrix membranes by the in situ fabrication of ZIF-8 nanoparticles (by varying the concentration of Zn(NO_3_)_2_ from 0.01 to 0.09 mol/L to give ZIF-8 concentrations of 12.2 wt% to 20.4 wt%) in polydimethylsiloxane (PDMS) matrix on PVDF support. A schematic representation of the fabrication process is shown in Figure 3. The resultant ZIF-8@MMM membranes were employed for the permselective pervaporation of ethanol from aqueous solution. It was found that ZIF was highly compatible and uniformly dispersed in the PDMS matrix, which enhanced the nucleation of ZIF-8 nanoparticles in the first place. However, moderate aggregation of ZIF-8 nanoparticles was also observed. ZIF-8 also enhanced the hydrophobicity, thermal stability and ethanol affinity of the produced ZIF-8@MMM membranes due to the nanoparticle’s super hydrophobicity and high adsorption capacity for ethanol which created preferential pathways for ethanol transfer and reduced the membrane permeation energy barriers for penetrants. Enhanced membrane hydrophobicity was proved by water contact angle tests which were found to increase from 111° for the PDMS/PVDF membrane to 138° for the ZIF-8@MMM after 10 min reaction time at a concentration of Zn(NO_3_)_2_ of 0.05 M. However, the contact angle of the ZIF-8@MMM membranes decreased to 127° when the reaction time was increased to 30 min. Ethanol affinity was also confirmed by contact angle measurements using ethanol. ZIF-8@MMM fabricated at 10 min reaction time also showed the lowest ethanol contact of 20° compared to the neat PDMS/PVDF of 45° and other modified membranes fabricated at different reaction times. The increased ethanol contact angle with time may be attributed to the decrease in membranes surface roughness. At 0.05 M concentration of the Zn(NO_3_)_2_ and reaction time of 10 min, ZIF-8@MMM showed anti-tradeoff behavior of increasing both the permeation flux and separation factor at an operating temperature of 40 °C, permeate pressure of 300 Pa, and 5 wt% ethanol aqueous solution. The anti-tradeoff phenomenon at the optimum reaction time of 10 min may be explained by the thicker active layer formed at higher reaction times as well as the precipitation of the ZIF-8 nanoparticles on the PVDF layer rather than the PDMS matrix. Whereas, increasing the Zn(NO_3_)_2_ concentration beyond 0.05 M increased the packing density of the hydrophobic ZIF-8 nanoparticles, so the water permeation flux decreased while the separation factor of ethanol increased due to the increase of ethanol preferential pathways [98].

Karimi et al. grew ZIF-8 particles on a PVDF porous membrane structure by a counter-diffusion method. The counter-diffusion method involves placing the membrane between two compartments, filled separately by Zn(NO_3_)_2_·H_2_O solution and 2-methyl imidazole solution. After a certain diffusion and crystallization time, ZIF-8 particles were found to form on the membrane structure due to the diffusion of Zn^2+^ and Hmim ions through the front and back pores of the membrane. The diffusion and crystallization time were varied and studied to define the optimum time for the best membrane performance. Energy Dispersive X-Ray (EDX) mapping results showed (Figure 4) that at a contact time of 5 h, a continuous and uniformly distributed layer of ZIF-8 nanocrystals covered the whole surface of the PVDF membrane. At lower contact times, the number of the nanocrystals formed was insufficient to cover the membrane due to the insufficient contact time. Whereas, increasing the contact time beyond 5 h led to the formation of a dense layer with large grain sizes. This may be attributed to the varying concentrations of Zn^2+^/Hmim ions. Membrane filtration performance tests showed that the water permeance increased in modified membranes that prepared at longer contact time and reached a maximum value for a modified membrane prepared at 5 h reaction time to a value of 134.56 L/m^2^∙h. This performance enhancement may be due to several reasons. First, the hydrophobic nature of the ZIF-8 particles that generates special channels with a hydrophobic nature decreases the friction between water molecules and the walls of these channels. Second, the water particle size (2.8 Å) is lower than the window size of the ZIF-8 particles (4–4.2 Å), so water molecules can easily pass through the ZIF-8 layer. On the other hand, modified membranes prepared at higher reaction time witnessed a decline in water permeance. This can be explained by the formation of the dense layer at longer contact times, which blocks the hydrophobic channels for water permeance and hence decreases the permeation flux. The antifouling properties were found to decrease with increasing the thickness of the ZIF-8 layer i.e., increased surface roughness. This finding is due to the increase of the regions where fouling can accumulate. Membrane rejection performance was studied by investigating the removal of two anionic organic dyes, direct yellow 12 (DY12) and reactive blue 21 (RB21). Compared to the neat PVDF membrane, the rejection of ZIF-8/PVDF composite membrane increased by 1.2 and 1.4 times for the RB21 and DY12, respectively. It was found the rejection performance depends on two factors. First, the molecular size of dyes (1.1 nm) was greater than the cavities of the ZIF-8 membranes, so they were easily blocked by the ZIF-8 layer. Second, the surface charges of the ZIF-8 modified membranes and dyes. Since both possess negative charges, the dye rejection can be attributed to repulsion effect between them. The reusability of the optimum modified membrane was also investigated using 0.5 g/L bovine serum albumin solution. Water flux decline by 36% was obtained after three consecutive cycles [100].

Basu et. al. prepared MOF-incorporated polymeric membranes for the removal of the common analgesic acetaminophen (or paracetamol). Two structures of the thin film composite (TFC) membranes were prepared by interfacial polymerization. The first was the in situ growth of the ZIF-8 in the support polysulfone (PSF) layer and the deposition of PA separation layer on top of the modified support (PSF/PA-ZIF-8). The second was the layer-by-layer (LBL) assembly of ZIF-8/PA on top of the PSF support (ZIF-8/PA-PSF) as shown in Figure 5. The LBL method involves the successive immersion of the substrate in solutions containing the metal salt and solutions of the organic ligands. After each cycle of deposition, the substrate is washed by an adequate solvent to remove any traces of unreacted compounds or any physico-sorbed components. AFM imaging of the ZIF-8/PA-PSF membrane confirmed that the ZIF-8 particles were perfectly wrapped by the PA polymer matrix and were well dispersed and hence enhanced the attachment of the ZIF-8 particles to the PA layer. It was also found that increasing the ZIF-8 concentration increased the water permeance up to 4 L/m^2^∙h∙bar. On the contrary, increasing the filler concentration resulted in decreasing MgSO_4_ rejection dramatically. This inverse relation between permeance and rejection can be explained by the formation of microvoids at the polymer–particle interface, which permit salts transfer and hence decrease rejection. Following the same trend, increasing the filler loading also decreased the acetaminophen from 46% of the neat PA/PSF membrane to 5% of the ZIF-8/PA-PSF that can be attributed to the membrane defects i.e., microvoids [29].

Guo et al. coated the inner and outer surface of macroporous polyvinylidene difluoride (PVDF) hollow fiber by a layer of ZIF-8 using gelatin-assisted growth technique. The produced ZIF-8/gelatin/PVDF was used for wastewater treatment to remove Rhodamine B dye. PVDF hollow fiber was chosen because it has higher area/volume ratio as well as higher packing density than the inorganic-supported membrane. While the gelatin-assisted technique was chosen to overcome the limitations of the organic solvents synthesis that hindered growth of MOFs at elevated temperature, thus enabling the growth of a uniform crack free ZIF-8 thin layer at room temperature. A schematic diagram that illustrates the synthesis procedure and the SEM imaging of the produced film in the inner and outer surfaces of the PVDF hollow fiber is shown in Figure 6. The optimum time for fabrication of a well inter-grown, uniform, continuous and dense layer of the ZIF-8/gelatin on both the inner and outer surfaces is 30 min. Significant separation performance for the separation of small dye molecules such as Rhodamine B molecules from water with permeance of 137 L/m^2^∙h∙bar and rejection up to 97.5% was obtained from the ZIF-8/gelatin/PVDF hollow fiber membrane [30].

Ragab et al. synthesized polytetraflurourethylene (PTFE) double layer membrane doped with ZIF-8 for the removal of micropollutants from water by adsorption. The synthesized doped membrane was applied to test the removal of progesterone (PGS) as an example of the micropollutant. To prepare the modified membrane, solvent evaporation technique was used. The PTFE membrane was immersed in solutions of different concentrations of ZIF-8 to synthesize PTFE membranes with different ZIF-8 loading up to 20 wt%. Agglomeration was noticed when the ZIF-8 concentration was increased beyond 20 wt%. It was found that addition of 10 wt% ZIF-8 to PTFE membrane increased the capacity of adsorption by about 40%. Moreover, the membrane water flux was increased and the membrane permeability was doubled (5.48 × 10^4^ L/m^2^∙h∙bar) compared to the neat PTFE membrane (2.93 × 104 L/m^2^∙h∙bar). This enhanced performance may be attributed to the enhanced membrane pore size distribution that increased the membrane effective surface area and the low affinity of the membrane pores to water molecules. Another contribution to the enhanced water flux is the presence of the N–H functional group on the ZIF-8 that results in hydrogen bonding with water molecules, and therefore facilitating the water transport through the membrane. The elevated membrane permeability, which is several of magnitudes higher than the nanofiltration (NF) system, adds to the advantages of this modified membrane. The specific energy consumption is reduced considerably due to the elimination of the high pressure pumping system and hence simplifies the process. After three regenerations using high strength PGS concentrations, the membrane kept 95% of its original efficiency, which adds to the system cost effectiveness [103]. Comparison between membranes performance with different loading concentration and the justification for the 10 wt% filler loading for all the tests carried out throughout the paper are missing.

Duan et al. incorporated the hydrophobic, thermally, hydrothermally and chemically stable ZIF-8 nanoparticles into a polyamide (PA) matrix of a thin film nanocomposite (TFN) membrane and used it for desalination [35]. The organic ligand present in the ZIF-8 enhanced the compatibility of this MOF with PA. Addition of 0.05% of ZIF-8 increased the water permeance significantly up to 88%. The highest loading value (0.4%) increased the permeance to 162% over the neat PA membrane while maintaining the same NaCl rejection of 98% using brackish water RO conditions. The increase in water permeability may be attributed to the facilitated transport of water molecules through the hydrophobic passages of the filler. Addition of ZIF-8 also increased the hydrophilicity of the membrane but decreased the crosslinking of the TFN surface [39]. Zhu et al. synthesized a TFN membrane by incorporation of ZIF-8 integrated with poly(sodium 4-styrenesulfonate) (mZIF) into a PA layer via interfacial polymerization on top of a hydrolyzed polyacrylonitrile (HPAN) support with the purpose of enhancing the water permeability. The obtained results showed that the integration of PSS into ZIF-8 did not alter its chemical structure but enhanced its dispersion and stability in aqueous solutions. Investigations of the membrane surface showed that increasing the filler loading, led to an increase in the membrane surface roughness, and consequently both membrane hydrophilicity and membrane water flux were increased (i.e., by increasing the membrane surface area). Addition of 0.1% *w*/*v* mZIF enhanced the membrane water permeability by more than 200% (14.9 L/m^2^·h·bar) compared to neat membranes (6.94 L/m^2^·h·bar). This enhanced membrane performance can be attributed to the mZIF pores and the interfacial voids between mZIF and PA meanwhile the high rejection values of Na_2_SO_4_ was maintained. However, increasing the filler loading to 0.2% *w*/*v* decreased the membrane permeability to 12 L/m^2^·h·bar. This decrease can be explained by aggregates formation due to the uneven distribution of the mZIF particles during the interfacial polymerization (IP) process and consequently the formation of partially compact surface structure. The modified membrane also scored ultra-high rejections of reactive dyes like reactive black 5 and reactive blue 2 as the rejection values were over 99% at room temperature and 4 bar operating pressure [104].

Aljundi investigated the effect of addition of ZIF-8 in the PA membrane on the improvement of membrane fouling-resistance. Results showed that the contact angle of the modified membrane was lower than the neat membrane by nearly half. This decrease in contact angle indicates the improvement in membrane surface hydrophilicity. During brackish water desalination tests (2000 ppm NaCl and at 15 bar operation pressure), the permeate flux increased up to 107% with increasing the filler loading up to 0.4% *w*/*v*. This increase in the membrane permeability may be due to the increased membrane hydrophilicity, which attracts more water and facilitates the passage of water molecule through the membrane. Further, the increasing in filler loading resulted in significant decreases in the membrane permeability that can be attributed to the migration of the ZIF-8 particles to the membrane surface that increased the hydrophobicity of the filled membranes as well as the salt polarization phenomenon where salts tend to accumulate on the feed side. Membrane modified with ZIF-8 at the optimum filler loading of 0.4% *w*/*v* showed higher permeate flux by 50% compared to the commercial Dow membrane (Dow-SW30HR) whereas it was lower than Dow-BW30 by 91.9% with the salt rejection value maintained at 99.4%. Under the same conditions, and with the addition of bovine serum albumin (BSA) (100 mg BSA/L) as model foulant, the fouling tendency of the ZIF-8 doped membrane was remarkably enhanced. This was proved by comparing the initial permeate flux and permeate flux after 4 h of continuous operation. It was found that the membrane permeate flux decreased by 53% and 13% for the neat and modified membranes, respectively. Hence, the flux losses were reduced by 75%. However, cleaning for 40 min with water was insufficient to restore the initial pure water flux which implies the presence of irreversible BSA fouling [105].

Low et al. incorporated ZIF-L nanoflakes into polyehersulfone (PES) and investigated the functional membrane properties of the produced MOFs-MMM. A SEM image of the leaf-like structure of the ZIF-L is shown in Figure 7. Results showed that the MOFs-MMM at filler loading of 0.5% ZIF-L witnessed an increase in the water flux by approximately 1.75 times (378 ± 10 L/m^2^∙h) compared to the neat membranes without compromising the molecular weight cut-off. This increase in water flux is attributed to the increased porosity of the modified membrane due to the leaf-like structure of ZIF-L. Increasing the filler content beyond 0.5% resulted in decreased membrane porosity, hence decreased water flux. Fouling resistance test was carried by bovine serum albumin (BSA) under constant flux operation on MOFs-MMM with the optimum filler loading of 0.5 ZIF-L. The modified membrane fouling resistance was enhanced by approximately 200% with more than 80% water flux recovery after three fouling cycles. Incorporation of ZIF-L nanoparticles increased the modified membrane hydrophilicity, decreased the surface roughness and lowered the zeta potential. These enhancements may be the reason for the increased membrane fouling resistance [106].

### 3.2. Materials of Institute Lavoisier (MIL)

MIL-n are trivalent metal based porous carboxylates such as chromium(III), vanadium(III) and iron(III) and continued to the p-elements such as aluminum(III) and gallium(III). They possess significantly large channels/cages and their topologies are similar to zeolites except the fact that MIL-n has different pore sizes, surface chemistry and density. These MOFs have open-framework structure with pore sizes and shapes strongly dependent upon the strong host–guest interactions. Another interesting phenomenon observed for MIL-n series is that they are very stable in water in contrast to other porous MOFs. MIL-101 is a subclass of MIL-n, which has huge pore sizes and surface areas as well as gigantic cell volume of approximately 702,000 Å^3^ [113,114]. MIL-125 was first reported by Dan-Hardi et al. in 2009. MIL-125 was synthesized using titanium(IV) isopropoxide as the metal cluster and 1,4-benzene dicarboxylic acid as the organic linker. MIL-125 is thermally stable up to 290 °C, highly porous and has high surface area that allows the adsorption of organic molecules [115]. Figure 8 shows the structure of MIL-125.

Xu et al. filled a dense selective layer of polyamide (PA) on polystyrene (PS) support with MIL-101(Cr) and employed the produced TFN membrane for water desalination by RO. A schematic representation of the RO system used to test the performance of the prepared membrane is shown in Figure 9. MIL-101(Cr) is a hydrophilic chromium based porous MOFs material that has larger pore size and surface area compared to other water stable MOFs like ZIF-8 and UiO-66. The hydrophilic property of MIL-101(Cr) attracts more water molecule, which enhances the hydrophilicity of the membrane where it is doped. As the concentration of the doped nanoparticles increases, the contact angle of the produced TFN membrane decreases indicating the increased hydrophobicity of the produced membrane. On the other hand, increasing the filler concentration increases the surface roughness that might increase the decline in its contact angle. Stability tests were carried out at 16 bar and 25 °C for 50 h at 2000 ppm salt concentration on membranes containing different MIL-101(Cr) loadings. These results are summarized in Figure 10. MIL-101(Cr) porous structures create preferential direct pathways for the rapid transport of water molecules through the dense PA layer, which increases the water permeation by 44% at filler loading of 0.05% *w*/*v* over the neat PA membrane at NaCl salt concentration of 2000 ppm. But, the NaCl rejection is maintained at a value greater than 99%. Increasing the filler loading beyond 0.05% *w*/*v* increases water permeation but the salt rejection was dramatically decreased and can be explained due to the interfacial defects between the PA layer and the MIL-101(Cr) inner voids aggregation. This may be due to the good compatibility between the doped MIL-101(Cr) and the PA polymer matrix. The matching sizes of MIL-101(Cr) nanoparticles also support the PA layer to resist the compaction and rearrangement of the polymer chains due to the applied pressure during the RO process and maintain a high water permeation value at elevated concentrations of MIL-101(Cr) [40].

Ma et al. enhanced the decontamination of multivalent cations by synthesizing a positively charged NF membrane made up by incorporating NH_2_-MIL-101(Al) and NH_2_-MIL-101(Cr) into a chitosan polymer matrix. The modification of the MIL-101(Al) with the introduction of the NH_2_-group into its structure enhanced the dispersion of the prepared MOFs in the organic (chitosan) phase. The morphology of the produced MOFs was found to have a significant effect on the membrane NF performance. At the same filler loading of 15 wt% and salt concentration of 2000 ppm, the rod-like NH_2_-MIL-101(Al) possessed a higher flux than the grainy NH_2_-MIL-101(Cr) by 200% with the same salt rejection. Therefore, the NH_2_-MIL-101(Al) filler was chosen for further investigation of the MOFs/chitosan NF membrane performance. The salt rejection order was found to be MgCl_2_ > CaCl_2_ > NaCl > Na_2_SO_4_ and the highest rejection value was recorded for MgCl_2,_ which was up to 93.0% at the optimum filler loading of 15 wt%. This performance can be explained by the synergistic effect of size exclusion and electrostatic interactions [116].

### 3.3. University of Oslo (UiO-66)

UiO-66 belongs to the zirconium–carboxylate based MOFs family that possess substantial chemical stability in organic solvents; its stability in water is excellent compared to other classes of MOFs and it has exceptional thermal stability up to 550 °C. These substantial stabilities may be attributed to the compact structure of UiO-66 as well as the strong Zr–O bonds. UiO-66 pore sizes also provide preferential water pathways that enhance water passage while eliminating the passage of hydrated cations [108,109,117]. The structure of UiO-66 is shown in Figure 11.

Ma et al. doped UiO-66 into a PA selective layer to synthesize a TFN membrane and its separation properties and its performance as a forward osmosis (FO) membrane was investigated. The particles of UiO-66 were perfectly covered by the PA phase at lower concentration, whereas when the filler concentration was increased to 0.2 wt%, the UiO-66 particles migrated to the surface of the PA layer and the particles was partially exposed without any covering and without any aggregations. Increasing the filler loading also increased the hydrophilicity of the membrane, the result that was confirmed by contact angle measurements gave a contact angle of 23.9° for the 0.2wt% UiO-66 concentration. This very low contact angle may be attributed to the super hydrophilic nature of the UiO-66 particles. Incorporating 0.1 wt% of UiO-66 increased the water permeability from 2.9 to 3.33 LMH/bar at a minimal compromise of salt rejection (from 95.5% to 95.3% rejection). Increasing the filler concentration beyond 0.1 wt% resulted in a thicker rejection layer, which decreased the water permeability significantly. The 0.1 wt% loading also gave the highest water flux during the FO performance test by 25% than the neat membrane when deionized water was used as the feed and 2 M NaCl solution as the draw solution [117].

Liu et al. incorporated UiO-66 into a PA layer to synthesize a TFN RO membrane for boron removal. During brackish water desalination test, TFN increased both the salt rejection and the water flux with increasing filler concentration. When the optimum concentration of 0.05% *w*/*v* was reached, a significant increase in the water flux was observed (56.83 L/m^2^·h) over the neat membranes (36.76 L/m^2^∙h). On the other hand, the salt rejection was very slightly changed (only 0.27% increased rejection). This increase in water flux can be explained by the creation of preferential pathways for water passage due to enhanced porosity. Increasing the filler loading beyond the optimum concentration led to reverse results of decreased flux and salt rejection. This decline in performance may be attributed to the increase in the resistance to mass transfer through the formed thicker PA layer and to the formation of aggregates, respectively. In seawater desalination tests, the water flux was enhanced by 19% for TFN membranes containing 0.05% *w*/*v* over neat membrane while the salt rejection remained unchanged, whereas the boron rejection value was enhanced by 11% compared to the neat membrane. This enhancement can be attributed to the strong adsorption capacity for boron of the UiO-66 (11 mg/g) during short periods of time and due to its extraordinarily high surface area of 1121.6 (m^2^/g) [119].

### 3.4. Pore Forming MOFs

F300 is an iron-based MOF (iron benzene-1,3,5-tricarboxylate), which is less water soluble than A100—an aluminium based MOF (aluminium terephthalate), and C300—a copper based MOF (copper benzene-1,3,5-tricarboxylate) [96]. F300 (also named FE-BTC) is a crystalline solid material with 1,3,5-benzenetricarboxylic acid as the organic linker [120], a pore size of 22 Å and specific surface area of 1500 m^2^/g. F300 was employed in the separation of small organic compounds in the aqueous phase and it has high catalytic activity for a wide variety of Lewis acidity reactions [121]. A100, which has properties similar to MIL-53, is formed by the coordination between Al^3+^ and the nodal metal atom and 1,4-benzenedicarboxylic acid. Several studies performed on A100 confirmed that it fits the gas separation applications properly even at very low CH_4_ content in presence of N_2_ [96]. The metal constituent of C300 is the Cu atom while the organic linker is trimesic acid. C300 exhibits a Langmuir surface area of 2000 m^2^/g with particle size of 20–100 µm and pore diameter of 3.5–9 Å. C300 (Cu-BTC) was studied first by Chui et al. in 1999 and was extensively investigated for gas adsorption and storage [122,123].

Lee et al. embedded F300, A100 and C300 MOFs in a polyacrylonitrile (PAN) polymer matrix as a green template for the production of water treatment pressure-driven membranes with enhanced porosity and interconnectivity. The green template MOFs are water-soluble but solvent stable, so by placing the whole structure (polymer loaded with MOFs) in water, the latter dissolves leaving behind pores and forming a porous matrix membrane (PMM). This porous matrix results in improving the membrane permeability coefficient, which is determined by the membrane porosity, pore size and tortuosity. The membrane permeability order was C300 > A100 > F300 in which C300 gave the highest water permeability of 260.5 L/m^2^·h, whereas PMM synthesized using F300 had the lowest permeability due to the greater water stability of F300 that led to the partial removal of F300 from the PAN matrix. However, the membrane rejection remained unchanged before and after the addition of MOFs (i.e., membrane selectivity did not change). The increase in pore connectivity may be another reason for the enhanced membrane permeability [96]. Lee et al. also investigated the use of MOFs-based PMM formed by LBL self-assembly to improve the mass transfer efficiency in FO. F300, A100 and C300 were used to prepare the partially removed F300 PMM_F300_ and the totally removed A100 PMM_A100_ and C300 PMM_C300_, respectively and compare their performance as FO membranes. The contact angle of the MOF-based PMM enhanced significantly compared to the neat PAN membrane and the commercial FO membranes (30°, 44° and 62°, respectively) while the membrane thickness was not changed in a noticeable pattern. Compared to the neat PAN membrane, MOFs-based membranes had enhanced bulk porosity (76%–85% compared to 70% for neat membrane). PMM_C300_ showed the highest water permeability that was 63% higher than the neat membrane; this may be attributed to the highest membrane bulk porosity that PMM_C300_ possesses. This high bulk porosity offered facilitated pathways for the water permeation whereas a slight increase in the salt rejection was observed. The optimum water flux of 132 L/m^2^·h was also obtained for the PMM_C300_ when deionized water (DI) was used as the feed solution and 3.0 M MgCl_2_ was used as the draw solution due to the formation of macropores that increased the mass transfer efficiency [97].

Arjmandi et al. synthesized thin film PMM (TF-PMM) using MOFs as pore formers and investigated the resultant TF-PMM for FO application. Upon FO desalination tests using the TF-PMM, Caspeian water desalination gave a water flux of 117 L/m^2^∙h; but the water flux during orange juice concentration tests was 98 L/m^2^∙h. When DI water was used as the feed solution, the water flux was up to 141 L/m^2^∙h. These enhancements in water flux is not only attributed to the pores formed by the washed away MOFs, but also due to the opened polymer chains that offered additional pathways for the passage of species. Among all the studied parameters i.e., polymer solution concentration, drying temperature, MOFs particle sizes, the MOFs loading was the only significant parameter to affect the membrane porosity. Membranes with lowest filler loading had the lowest porosity and vice versa. Membrane cleaning results showed that the produced PMM has great potential to be reused with continuous cleaning processes [124].

## 4. Effect of MOFs Incorporation Technique on the Membrane Performance

Zhao et al. investigated the effect of incorporating MIL-53(Al), NH_2_–UiO-66 and ZIF-8 in a PA layer to synthesize a NF membrane [104]. The TFN membranes were synthesized by two different techniques namely blending (BL) and preloading (PL) interfacial polymerization on polysulfone (PS) support. The main difference between the two techniques is the method of incorporating MOFs into the polymer matrix. In the PL technique, MOFs are loaded directly onto the active layer (PS layer) before the interfacial polymerization (IP) process, while the BL method involves the pre-dispersing of the nanoparticles and trimesoyl chloride (TMC) in n-hexane then loading to the PS support. Water permeability tests as well as sugar (xylose) and salt rejection tests were carried out to evaluate the effect of adding different MOFs on the membrane filtration performance. TFN membranes performance was found to be highly affected by the properties of the MOFs as well as the preparation method and the filler loading. Due to the incorporation of MOFs with the TMC during the IP process in the BL technique, a low cross-linked PA layer was obtained which exhibits larger pore sizes and higher amounts of –COOH groups. The presence of large quantities of –COOH groups as a result of low degree of cross-linking, increased the negative charge density on the membrane surface, hence increased the NaCl rejection via electrostatic repulsion for filler loadings less than 0.15%. On the other hand, low cross-linking degree resulted in larger pore size membranes that increased the water permeability but lowered the sugar rejection. Although the hydrophilic nature of the incorporated MOFs (i.e., MIL-53(Al) and NH_2_–UiO-66) enhanced the miscibility of the aqueous and organic phases, the sugar rejection of the 0.1% of the latter MOFs was not enhanced compared to the neat membrane. The xylose rejection by the TFN membranes may be attributed to the size exclusion of the xylose molecule (0.73 nm) by the MIL-53(Al), NH_2_–UiO-66 and ZIF-8 MOFs of window sizes 0.86, 0.60 and 0.34 nm, respectively. However, water molecules of 0.28 nm size could not penetrate through the ZIF-8 particles due to the hydrophobic nature of ZIF-8, so water molecules preferred to transfer through the polymer matrix instead. When MOFs were directly deposited on the support surface in the PL method, the degree of cross-linking was not much affected by the presence of MOFs compared to the membrane prepared by the BL technique. On the basis of NaCl rejection, it was found that there are no distinct variations between the MOF-incorporated TFN membranes and the neat PA membrane. The major effect of MOFs incorporation in this case is the altering of the PA layer morphology and consequently increasing the available surface for water permeability. The optimum MOFs loading was 0.1% and 7.6 µg/cm^2^ that gave the optimum water permeability without sacrificing rejection for both synthesis techniques. The hydrophobic nature of ZIF-8 prohibited water permeation through the TFN membranes, hence the permselectivity enhancement due to ZIF-8 addition was negligible. In comparison to NH_2_–UiO-66, MIL-53(Al) binds more tightly to the PA layer which decreases the formation of nonselective voids and enhances the permselective properties of the synthesized TFN membrane filled with MIL-53(Al). TFN membrane synthesized by BL technique at MIL-53(Al) loading of 0.1% exhibits enhanced water permeability up to 7.2 LMH/bar which is 30% higher compared to corresponding TFC membranes while maintaining rejection of NaCl higher than 40% and xylose rejection higher than 65% [125]. In another study, Kadhom et al. evaluated the desalination properties of a TFN membrane prepared from PA doped with MIL-125 (titanium based MOFs) and UiO-66 at different filler loadings ranging from 0 wt% to 0.3 wt% as shown in Figure 12. Compared to UiO-66, MIL-125 had a poorer dispersion into the polymer matrix and a lower hydrophilic effect. This may be attributed to the more organic ligands present in MIL-125 structure compared to UiO-66 that masked the metal ion. Increasing the filler loading beyond 0.3 wt% may form aggregates that can cause cracks in the TFN membrane, and consequently block the passage of water molecules and form voids that allows saline water to transfer through the membrane, eventually leading to decreased water flux and salt rejection. To test the TFN membrane desalination performance, a salt concentration of 2000 ppm was used at a trans-membrane pressure of 3000 psi. For MIL-125, the water permeability increased significantly from 62.5 L/m^2^·h (of the neat PA membrane) to 85.0 L/m^2^·h with filler loading of 0.3 wt% whereas the salt rejection was only enhanced by 0.2%. On the other hand, addition of only 0.15 wt% of UiO-66 resulted in an increase in water permeability to 74.9 L/m^2^·h but the salt rejection was very limited as it only increased from 98.4% to 98.8% [42].

## 5. MOFs-MMMs Challenges, Solutions and Future Prospective

MOF-based MMMs are promising candidates to solve the trade-off issue between the permeability and selectivity of the polymeric membranes, where membranes with high permeability suffer from low selectivity and vice versa. However, the practical results obtained differ significantly from the theoretical expected results obtained from simulation data [126].

Although more than 20,000 MOFs were developed in the last ten years, very few of them were used to synthesis MOF-based MMMs. This is due to several reasons, some of which are attributed to the characteristics and application of the produced MMMs which include stability, pore size, selectivity and diffusivity, whereas the others are attributed to the MOFs–polymer interactions like the presence of interfacial defects and their impact on performance [126]. Other than these previously mentioned MMMs issues, the main challenges facing the MOFs-MMMs with the targeted separation performance and desired membrane characteristics are: (1) selection of a compatible MOFs/polymer system, which is associated with the optimum performance for a certain morphology, (2) MOFs low water stability compared to other fillers such as zeolites and silica gel, (3) MOFs (inorganic phase) are usually not well dispersed in the polymer (organic) matrix and may form aggregates at higher concentrations (more than 10%), (4) improper interfacial interactions of MOFs/polymer systems may lead to defects in the produced MOFs-MMMs such as filler’s pores blocking, rigidification of the polymer around the filler particles and interfacial voids [97,119,126], (5) the difficulty of scalability of the MOFs-based MMMs due to the insufficient adhesion between the MOFs as fillers and the polymer matrix, (6) stability of MOFs in the used fluid (e.g., water or salt solution) is a steering tool that affects the stability and sustainability of the performance of the produced MOFs-MMMs.

Scaling up of MOFs synthesis to an industrial and commercial scale is becoming a more attractive trend recently. This may be attributed to the very limited real life applications of MOFs-MMMs, although there are some MOFs that are readily commercially available. The reason for this limited applications field is the use of hazardous reactants during MOFs synthesis and the difficult reaction conditions that limit their synthesis in pilot plants or at industrial scale [95].

Many approaches were suggested to address these previous challenges. A very useful tool to select the most appropriate MOFs–polymer combination for a definite separation application is carrying out surveys by computational methods on the topological properties of the selected species [127]. Relation evaluation between molecular-level properties of MOFs and the MMMs performance and application is also another step on the road for selecting the most compatible MOFs for a certain polymer matrix. The polymer matrix choice itself is a point of equal importance [128]. On the other hand, homogeneous MOFs dispersion in the polymer matrix can be achieved by reducing the filler particle size that will have more affinity with the polymeric matrix, hence improving the MOFs-MMMs performance. Decreasing the particle size will also increase the polymer/MOFs interface area, creating more selective pathways for species permeation. Surface modification is also another technique to improve the adhesion of MOFs and the polymer. Surface functionalization can be carried out by introducing the appropriate functional group to the MOFs surface that is compatible with the polymer matrix [126].

Some of the future aspects that can be considered for further enhancement of the MOFs-MMMs are: (1) Investigation of new synthesis techniques to manufacture MOFs-MMMs with oriented MOFs. Oriented MOFs will offer more facile pathways for the permeation of species from one side of the membrane to the other, which is an important factor in processes like MD and pervaporation. (2) Incorporation of MOFs on the membrane surface without sacrificing their good adhesion to the polymer matrix in cases where the surface properties need to be more pronounced. (3) Conventional synthesis techniques lead to the perfect encapsulation of MOFs within the polymer matrix, hence decreasing the available surface area of MOFs for separation. Therefore, novel techniques introduced to bind the MOFs particles at a distance from the membrane surface may be the answer to increase the available MOFs surface area. (4) New synthesis routes at mild conditions and using non-hazardous materials should be investigated to solve the scaling up issue of the MOFs-MMMs.

## 6. Conclusions

Since the manufacture of the first membrane, many efforts were spent on enhancing the membrane properties whether thermal, mechanical or fouling resistance. MOFs are a class of hybrid materials made up of organic ligands and metal clusters that possess huge surface areas and exceptional properties. MOFs-based MMMs were introduced to solve different membrane issues as well as the trade-off issue between selectivity and permeability of polymeric membranes. MOFs organic ligands increased MOFs compatibility with the polymeric matrices, hence well dispersion of the MOFs fillers is achieved. However, increasing MOFs loading beyond 10% in most cases led to formation of aggregates that had a significant effect on the membrane performance. The MOFs aggregates increased the membrane pore sizes, consequently forming non-selective voids which increased water and salt permeation, hence decreased the salt rejection. MOFs structure and properties and the way the filler is incorporated in the polymeric matrix also affected the MOFs-MMMs properties and performance. On one hand, membrane hydrophobicity/hydrophilicity is dependent on the MOFs itself whether it is hydrophobic or hydrophilic in nature and if it was perfectly encapsulated by the membrane matrix or deposited/migrated to the surface. Water preferred to be transported through hydrophilic MOFs such as MILs and UiO-66. On the contrary water was transported through preferential path ways around hydrophobic MOFs such as ZIFs. On the other hand, geometry (window size) affected the water permeation and salt passage, where species with smaller molecular size than the MOFs window size can pass through the MOFs structure and those with bigger molecular size are rejected. Different synthesis techniques were investigated for the manufacture of MOFs- based MMMs. For example, an in situ growth technique produced MOFs-MMMs with better dispersion and compatibility, while gelatin-assisted seed growth produced a uniform crack free thin film MOFs layer at room temperature. Despite all the merits of these hybrid membranes, it has its own complexities and difficulties that restrict its large-scale application and fabrication. Some of which are the high cost, difficult and dangerous MOFs synthesis techniques and stability and sustainability of performance. Additionally, the use of MOFs in membranes used in water purification applications may be considered as a potential environmental and health hazard. Therefore, new synthesis techniques with mild conditions and utilizing nonhazardous compounds, selection of a compatible MOF/polymer system and the specific properties that need to be promoted for a definite application should be well investigated. To conclude, MOFs-MMMs have a great potential in different separation applications due the exceptional properties offered by MOFs but its success, competitiveness and upscaling needs further persistent efforts to solve problems identified with their fabrication and application.

## Figures and Tables

**Figure 1 membranes-09-00088-f001:**
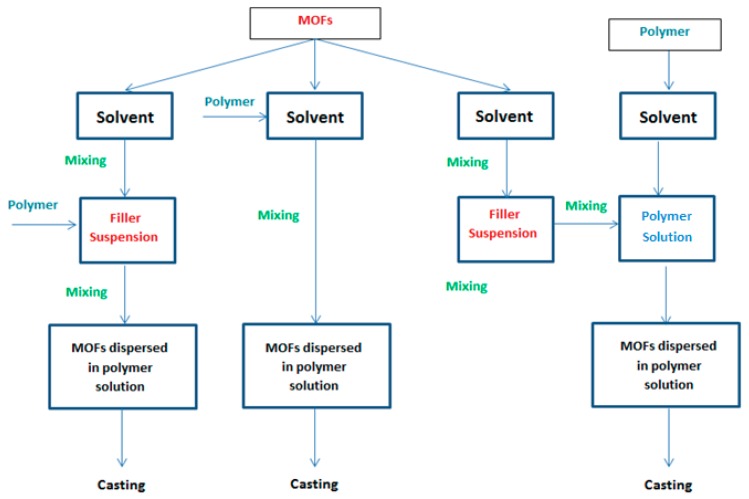
Schematic diagram of the different routes followed in the blending method to synthesis MOF_S_-MMMs.

**Figure 2 membranes-09-00088-f002:**
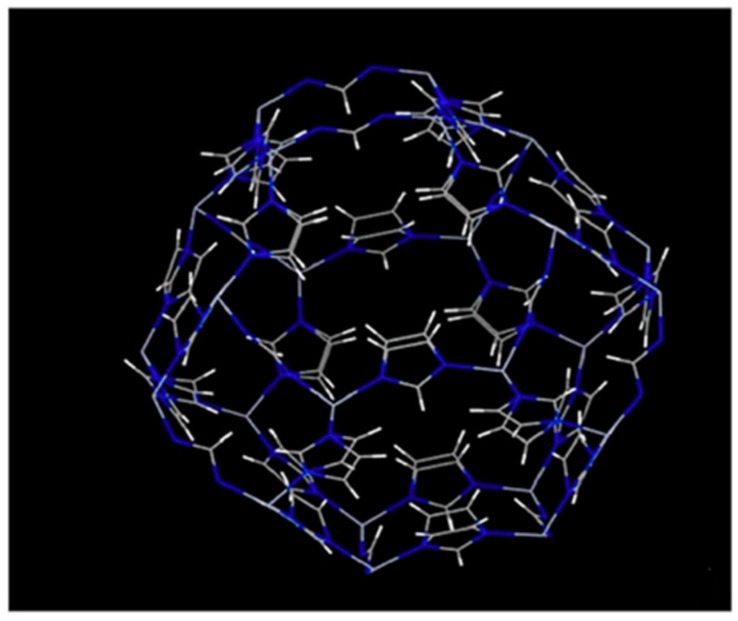
Simulated structure of zeolite imidazole framework (ZIF)-8 [38].

**Figure 3 membranes-09-00088-f003:**
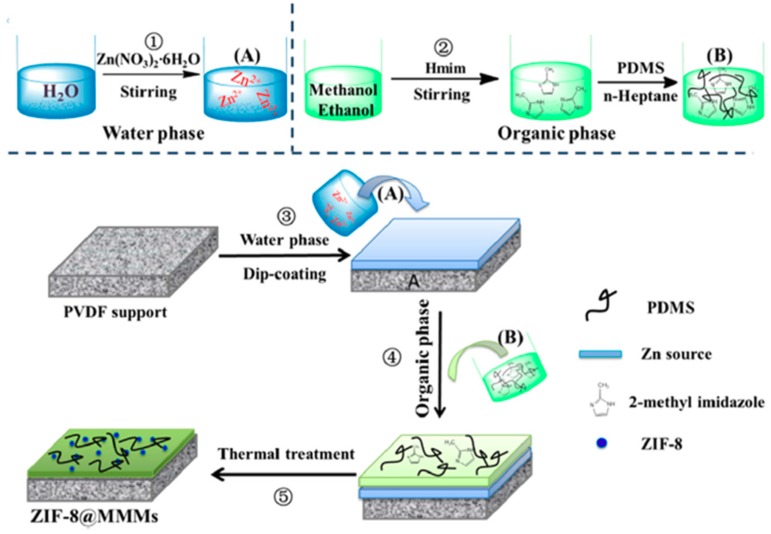
In situ fabrication procedure of the ZIF-8-MMM [98].

**Figure 4 membranes-09-00088-f004:**
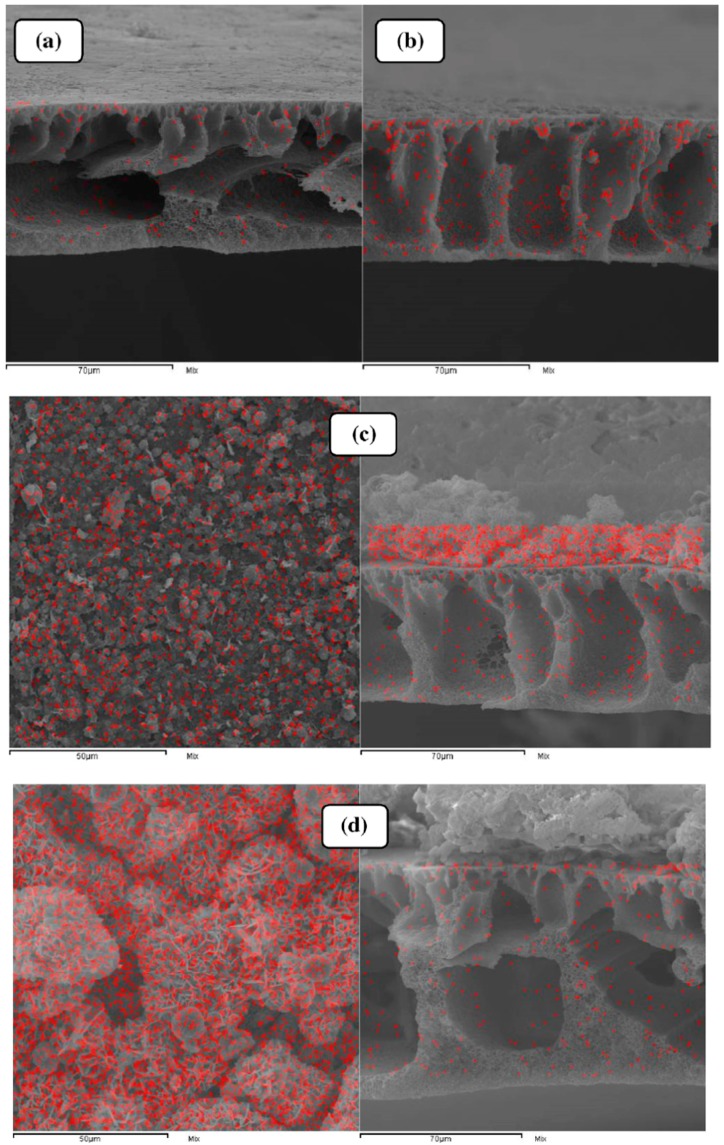
Energy Dispersive X-Ray (EDX) mapping of Zn element in the cross-section of the ZIF-8/PVDF composite membranes prepared at different contact times of (**a**) 1 h and (**b**) 3 h; EDX mapping of Zn element on the surface and cross section of the ZIF-8/PVDF composite membranes prepared at contact times of (**c**) 5 h, (**d**) 8 h and (**e**) 24 h [100].

**Figure 5 membranes-09-00088-f005:**
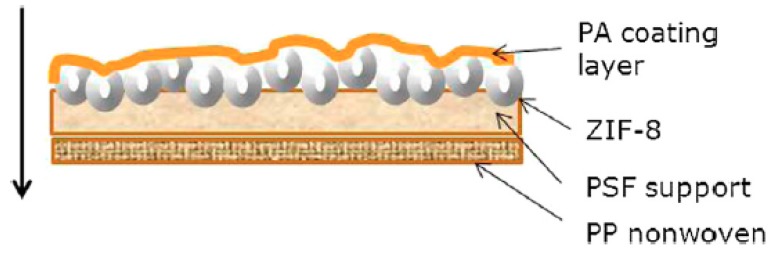
Schematic diagram representing the layer-bay-layer (LBL) technique for the preparation of ZIF-8-MMM [29].

**Figure 6 membranes-09-00088-f006:**
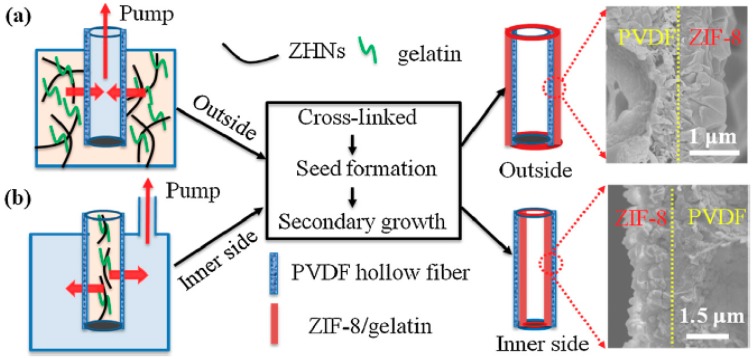
Schematic diagram of gelatin-assisted growth of ZIF-8 on the (**a**) outer and (**b**) inner surface of a PVDF hollow fiber with their corresponding SEM images [30].

**Figure 7 membranes-09-00088-f007:**
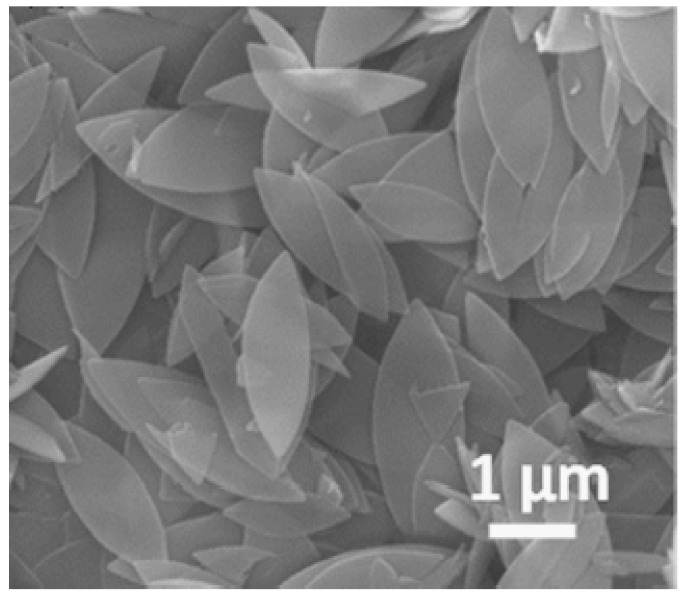
SEM image of the ZIF-L leaf-like nanoflakes [106].

**Figure 8 membranes-09-00088-f008:**
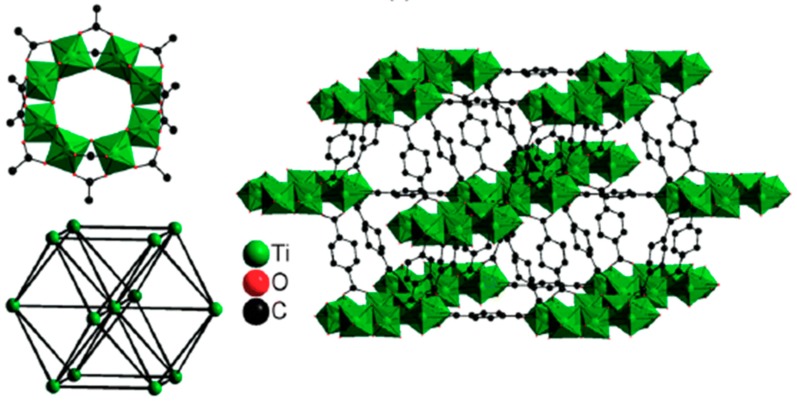
Materials of Institute Lavoisier (MIL)-125 structure [107].

**Figure 9 membranes-09-00088-f009:**
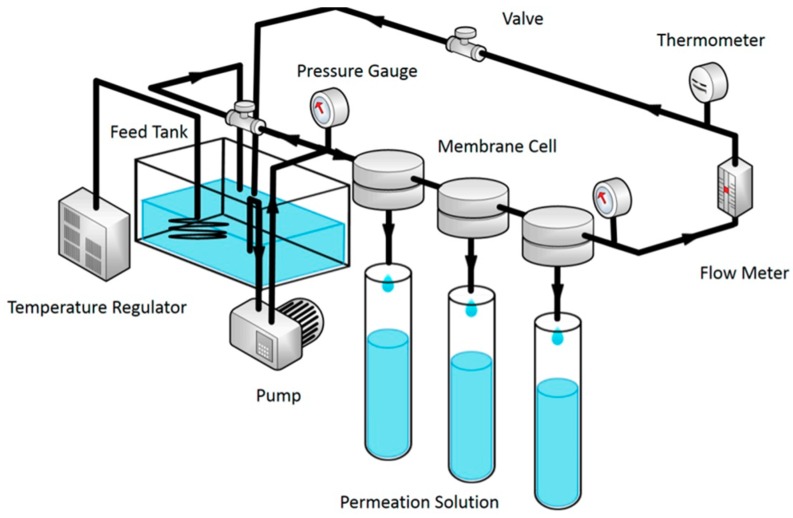
A schematic representation of the reverse osmosis (RO) system used to test the thin film nanocomposite (TFN) membrane doped with MIL-101(Cr) [40].

**Figure 10 membranes-09-00088-f010:**
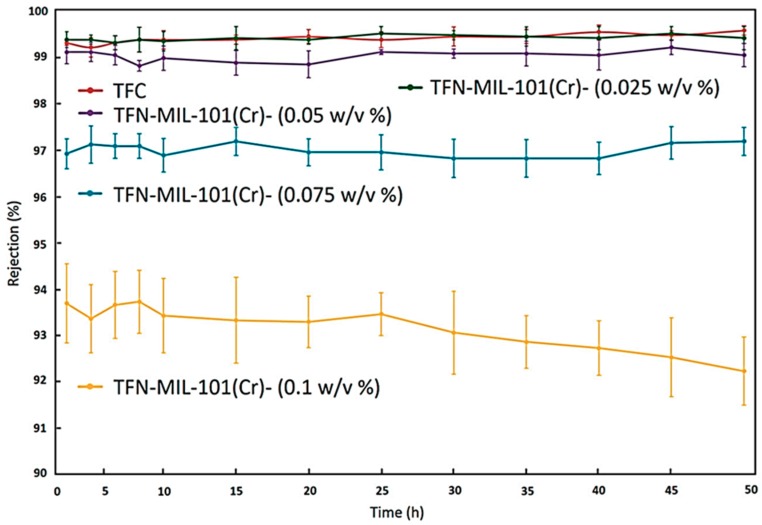
Stability test results of the TFN filled with MIL-101(Cr) at 2000 ppm NaCl aqueous solution at 16 bar and 25 °C for 50 h [40].

**Figure 11 membranes-09-00088-f011:**
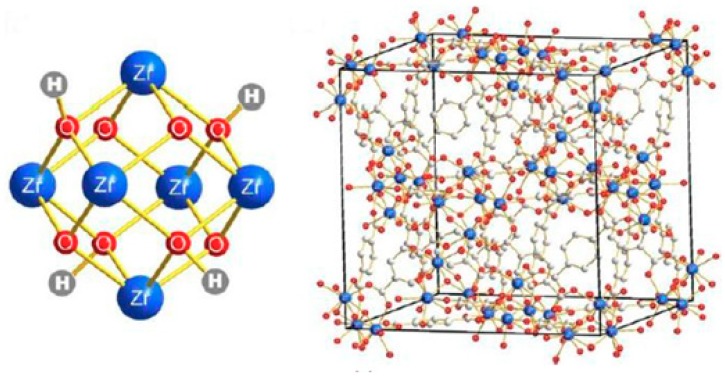
University of Oslo (UiO)-66 structure [118].

**Figure 12 membranes-09-00088-f012:**
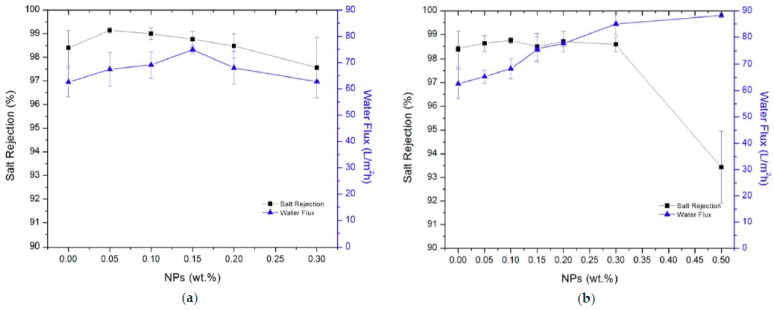
TFN membrane desalination performance filled with (**a**) UiO-66 and (**b**) MIL-125 [42].

**Table 1 membranes-09-00088-t001:** Different synthesis techniques of metal–organic frameworks-mixed metal membranes (MOFs-MMMs).

Blending	In Situ Growth	Layer-By-Layer	Gelatin-Assisted Seed Growth
The blending technique is divided into three methods: Dispersion of the already prepared filler in a solvent, then mixing the polymer to that suspension before casting. Dissolving the polymer in a proper solvent, then the MOFs are added and casting is carried out afterwards. The filler and the polymer are dispersed and dissolved in solvents, separately. Then, the filler suspension and polymer solution are mixed together before casting.	In this process, MOFs are produced by covalent coordination between the metal clusters and the organic ligand together with the membrane formation or within the pores of an already prepared membrane structure, which result in better dispersion and compatibility of the produced MOFs in the polymer matrix.	The LBL method involves the successive immersion of the substrate in solutions containing the metal salt and solutions of the organic ligands. After each cycle of deposition, the substrate is washed by an adequate solvent to remove any traces of unreacted compounds or any physico-sorbed components. Hence a layer of well-intergrown continuous dense film of the targeted MOFs is created on the substrate surface.	The substrate is immersed in a gelatin solution containing the MOFs seeds. This method was developed to overcome the limitations of the organic solvents synthesis that hindered growth of MOFs at elevated temperature thus enabling the growth of a uniform crack free MOFs thin layer at room temperature.

**Table 2 membranes-09-00088-t002:** Properties and fabrication conditions of some successful prepared MOFS-mixed matrix membranes.

Type of Filler	Polymer Matrix	Support	Composite Membrane Fabrication Technique	Optimum Conditions	Permeation Flux	Separation/Rejection Factor	Filler Loading/Particle Size	Selection Criteria	Application Process	Reference
ZIF-8	PDMS	Polyvinylidene fluoride (PVDF)	In situ fabrication (growth) of MOFs within the polymer matrix.	Time: 10 min Conc.: 0.05 M of Zn(NO_3_)_2_	1868 g/m^2^·h	Ethanol separation factor 12.1	12.2–20.4 wt% based on starting Zn(NO_3_)_2_ concentration of 0.01–0.09 M	Hydrophobicity Increased thermal stability of modified membrane Increased affinity for ethanol	Pervaporation	[98]
ZIF-8	PA	PSF	Deposition of ZIF-8 particles dispersed in m-Phenylene diamine (MPD) solution on the microporous support prior to interfacial polymerization of the PA layer.	Mean particle size ZIF-8 (150 nm) at filler loading of 0.2 wt%/vol%	3.95 L/m^2^·h·bar	NaCl rejection 99.2%	0.2 wt% and 0.4 vol% 60, 150 and 250 nm	High water permeability Small window sizes Good water stability High specific surface area	RO	[99]
ZIF-8/chitosan	PVDF		PVDF membrane was immersed in a coating solution of ZIF-8 particles, chitosan, PEG and DI water.		137 L/m^2^·h	rejection up to 97.5%		The gelatin-assisted technique was chosen to overcome the limitations of the organic solvent synthesis of the hindered growth of MOFs at elevated temperature that enabled the growth of a uniform crack-free ZIF-8 thin layer at room temperature.	Removal of Rhodamine B dye	[38]
ZIF-8	PA	PSF	ZIF-8 particles were dispersed in TMC/hexane solution used in the IP.				0.05% *w*/*v* to 0.40% *w*/*v*	Theoretically faster water transport within the framework Better compatibility with the PA matrix	Desalination by RO	[39]
ZIF-8	Porous (PVDF)		Contra-diffusion synthesis method was used to create a uniform layer of zeolitic imidazolate framework- 8 (ZIF-8) on the porous polyvinylidene fluoride.	5 h contra-diffusion synthesis time	134 L/m^2^·h	98.32% for reactive blue 21 dye and 82.25% for direct yellow 19 dye	Continuous layer	Well defined cavities Accessible window sizes The hydrophobic nature of ZIF-8 accelerates the passage of water molecules due to the small resistance of the ZIF-8 walls and water molecules	Dye removal	[100]
ZIF-8	PAN	PSS	Coordination-driven in situ self-assembly for the synthesis of hybrid ZIF-8/PSS membrane on the surface of a polyacrylonitrile (PAN) support.	Starting solution of 0.05 M concentration of Zn(NO_3_)_2_	265 L/m^2^·h·MPa	98.6% of MB dye	Uniformly dispersed layer on the membrane— 150 nm at 0.05 M Zn(NO_3_)_2_	High dye-retention rate High flux of the produced modified membrane	Nanfiltration of MB dye from water	[101]
ZIF-8	PVP/PES		Blending of previously prepared ZIF-8 particles with the polymer matrix.			99.6% dye removal at 3% filler loading	1–3% ˂100 nm	High separation ability of ZIF-8 particles due to its zeolite like structure	Malachite green dye removal in a cross-flow system	[102]
ZIF-8	PA	PSF	Two different membrane structures were obtained by in situ growth of ZIF-8 particles in the PSF support then followed by deposition of a PA separation layer on top of the modified membrane. LBL assembly of ZIF-8/PA on top of PSF support.	4 L/m^2^·h·bar			0.02, 0.04, 0.06, 0.08, 0.1 g/100 mL	ZIF-8 significant separation ability	Separation of pharmaceuticals from aqueous streams	[29]
ZIF-8/Gelatin	PVDF hollow fiber		Gelatin-assisted growth technique.	30 min reaction time to produce well inter-grown, uniform, continuous and dense ZIF-8/gelatin layer	137 L/m^2^·h·bar	97.5% dye rejection	Uniform, continuous and dense ZIF-8/gelatin layer on the inner and outer surface of the PVDF hollow fiber	ZIF-8 thermal and chemical stability Enhanced hydrophobicity of modified membrane Enhanced surface porosity	Rhodamine B dye removal from waste water and AGMD	[30]
ZIF-8	PTFE		The modified membrane was prepared by solvent evaporation technique. The PTFE membrane was immersed in solutions of ZIF-8 of different concentrations to synthesize PTFE membranes with different ZIF-8 loading up to 20 wt%.	10 wt% ZIF-8 filler	5.48 × 10^4^ L/m^2^·h·bar	The capacity of adsorption was increased by about 40%	Different ZIF-8 loading up to 20 wt%	Thermal and chemical stability Simple preparation method Relatively low cost of raw materials Large surface area available for adsorption	Micropollutants removal (progesterone (PGS))	[103]
mZIF	PA	Hydrolyzed PAN	Modified ZIF particles were dispersed in the polypiperazine (PIP) phase used for the IP process.	Filler loading of 0.1% *w*/*v*	14.9 L/m^2^·h·bar	Rejection values were over 99%	0.05%, 0.10% and 0.20% *w*/*v*	Hydrothermal, thermal and chemical stability of ZIF-8 particles.	Reactive black 5 and reactive blue 2 dyes nanofiltration	[104]
ZIF-8	PA	PSF	ZIF-8 nanoparticles were dispersed in the TMC hexane solution used for the IP process.	Filler loading of 0.4% (*w*/*v*)	34.5 L/m^2^·h	99.4	0.1 wt%, 0.2 wt%, 0.4 wt%, 0.6 wt% and 0.8 wt%	Good compatibility with the polymeric matrix The hydrophobic nature of ZIF-8 accelerates the passage of water molecules due to the small resistance of the ZIF-8 walls and water molecules Enhanced salt rejection due to the synergistic of steric/Donnan exclusion	Desalination by RO	[105]
ZIF_L nanoflakes	PES		Non-solvent induced phase separation.	0.5% filler loading	378 ± 10 L/m^2^·h		0.25%, 0.5%, and 1% PES/ZIF-L	Improvement of the filler–polymer compatibility through the flake-shaped ZIF-L, hence enhanced membrane performance.	UF	[106]
MIL-101(Cr)	PA	PSF	MIL-101 (Cr) nanoparticles was added into a 0.1% *w/v* TMC hexane solution used in the IP process.	Filler loading of 0.05% *w/v*	2.25 L/m^2^·h·bar	˃99%	0.025% to 0.1% *w*/*v*	Improved and increased water channels due to MIL-101 (Cr) larger pore size and surface area Increased membrane surface hydrophilicity due to the hydrophilic nature of MIL-101 (Cr) Sustains the channels architecture during the RO high pressure operation	RO desalination	[40]
NH_2_-MIL-101(Al) NH_2_-MIL-101(Cr)	chitosan	PSF	Solvent casting of solution containing MOF particles were dispersed in chitosan on top of the PSF.	15 wt% filler loading	NH2-MIL-101(Al) possessed a higher flux than the grainy NH_2_-MIL-101(Cr) by 200% with the same salt rejection.	93% MgCl_2_ rejection	0%, 5%, 10%, 15% and 20%	High stability in water and common solvents High surface area High porosity	NF	[107]
UiO-66	PA	PSU	UiO-66 particles were dispersed in TMC/n-hexane phase of the IP process constituents.	0.1 wt% filler loading	3.33 L/m^2^·h·bar	95.3% salt rejection	0.05 wt%, 0.1 wt%, 0.15 wt% and 0.2 wt%; around 512 nm	The hydrophilicity and stability of UiO-66 make it suitable for incorporation in the PA layer. UiO-66 water stability triggers its use in aqueous operations. Facilitated water permeation through the well-defined sub-nanometer pores of UiO-66	FO	[108]
UiO-66	PA	PSF	UiO-66 particles were dispersed in TMC/n-hexane phase of the IP process constituents.	0.05% *w*/*v*	56.83 L/m^2^·h for BW desalination tests and 61.32 L/m^2^·h for the SW desalination tests	99.35% salt rejection for BW desalination tests and remained unchanged for the SW desalination tests 91.2% boron rejection	0.025%, 0.05%, 0.075% and 0.1% (*w*/*v*) UiO-66 Loading—50 nm	Longer pathways for diffusion and selective permeation of molecules through the tortuous channels of UiO-66 Chemisorption of boron increases the adsorption capacity significantly Good compatibility with PA The acid/alkali stability of UiO-66 allows chemical washing of the membranes	SW and BW desalination Boron removal	[109]
F300, A100 and C300	PAN		Casting of well dispersed MOFs in PAN phase.	0.1 wt%	Membrane doped with C300 scored the highest membrane permeability of 260.5 L/m^2^·h			Stable MOFs in polar organic phase but have very low water stability so easily dissolves in aqueous phase	PMM manufacture	[96]
F300, A100 and C300	PAN		Alternative immersion of the MOF based PMM in PSS and PAH solution for the target of fabricating rejection layer via LBL method.		Membrane doped with C300 scored the optimum membrane permeability of 132 L/m^2^·h			MOF particles incorporated as removable fillers to synthesize FO membranes with high porosity	PMM manufacture to be utilized in FO	[97]

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
