# Peer review of "Metal Organic Framework Based Polymer Mixed Matrix Membranes: Review on Applications in Water Purification"

_membranes, 2019, doi:10.3390/membranes9070088_

Reviewer 1 Report

The manuscript by Elrasheedy and co-workers reviews the field of MOF-based MMMs for water treatment. The literature is worthwhile to be summarized in this context. The content of the manuscript fits well in the scope of the journal. However, there are several issues that must be addressed before publishing the work. The main criticism is the fact that the review needs to be more than a simple compilation of literature. The detailed comments to improve the manuscript are given below.

1. The authors should provide some future perspective on the field of MOF-MMM. What’s next? What are the current challenges of the field that needs to be addressed? What are the potential solutions to explore?

2. It is necessary to provide a comprehensive table that compares the MOF-MMMs for aqueous applications. MOFs, polymer matrix, loadings, performance etc should be compared. 

3. The different membrane preparation techniques for MOF-MMMs should be summarized and compared. A figure with several panels that show schematics of these techniques should be included in the manuscript.

4. The following MOF-MMM for water applications are missing from the review and should be discussed (DOI: 10.1021/jacs.5b02276; 10.1039/C7TA00339K; 10.1021/acs.cgd.6b01398; 10.1021/acsanm.8b01902; 10.1021/acsanm.8b00501; 10.1038/srep03740; 10.1016/j.memsci.2019.02.020).

5. The manuscript lists most of the articles that fall within the scope but lacks critical assessment. The authors should add some critical comments as appropriate throughout the manuscript.

6. What are the main limitations of MOFs as fillers for MMMs? In depth discussions should be included in the manuscript. Stability issues and loss of performance as well as need for long-term performance studies should be mentioned.

7. Group large number of references should be avoided. These do not provide any useful information for the readers, e.g. [41-50].

8. The issues of scale-up of both MOFs and MOF-based membranes need to be mentioned.

9. The figures taken from the literature are of low resolution. The authors should either redraw them or source a higher resolution image.

10. The manuscript focuses on water application of MOFs. However, the authors should briefly mention in a sentence the potential and challenges of MOFs for non-aqueous MMMs and give some examples (DOI: 10.1039/C4TA00628C; 10.1039/C5RA10259F; 10.1016/j.memsci.2016.01.024).

11. The fabrication of green and sustainable membranes are becoming increasingly important. What is the potential of MOF-MMMs in that perspective? Are there any green MOFs? Have they been applied in any green process? These considerations should be added to the review.

12. The conclusion section is vague. More tangible information that guide the readers on structure-function-performance relationship should be provided. The most promising MOFs for aqueous applications should also be highlighted here.

Author Response

Dear Dr. Angela Yang,

Thank you very much for the constructive and interesting reviewers’ comments on submitted  manuscript that will certainly help to enhance the quality of the manuscript. I am pleased to inform you that the revised manuscript has been updated according to the reviewers’ comments and addressed each comment separately. The reviewers’ comments and answers are listed below.

Reviewer#1

The manuscript by Elrasheedy and co-workers reviews the field of MOF-based MMMs for water treatment. The literature is worthwhile to be summarized in this context. The content of the manuscript fits well in the scope of the journal. However, there are several issues that must be addressed before publishing the work. The main criticism is the fact that the review needs to be more than a simple compilation of literature. The detailed comments to improve the manuscript are given below.

1.   The authors should provide some future perspective on the field of MOF-MMM. What’s next? What are the current challenges of the field that needs to be addressed? What are the potential solutions to explore?.

In full agreement with the reviewer, a new section has been  added page

5. MOFs-MMMs challenges, solutions and future prospective

MOF-based MMMs are promising candidates to solve the trade-off issue between the permeability and selectivity of the polymeric membranes, where membranes with high permeability suffer from low selectivity and vice versa. However the practical results obtained differs significantly from the theoretical expected results obtained from simulation data [124].

Although more than 20,000 MOFs were developed in the last ten years, very few of them were used to synthesis MOF-based MMMs. This is due to several reasons, some of which are attributed to the characteristics and application of the produced MMM which include stability, pore size, selectivity and diffusivity, whereas the others are attributed to the MOFs-polymer interactions like the presence of interfacial defects and their impact on performance [124]. Other than that the main challenges facing the facing the MOFs-MMMs with the targeted separation performance and desired membrane characteristics are: (1) Selection of a compatible MOF/polymer system, which is associated with the optimum performance foe a certain morphology (2) MOFs low water stability if compared to other fillers such as zeolites and silica gel, (3) MOFs (inorganic phase) are usually not well dispersed in the polymer (organic) matrix and may form aggregates at higher concentrations (more than 10%), (4) Improper interfacial interactions of MOF/polymer systems may lead to defects in the produced MOFs-MMMs such as filler's pores blocking, rigidification of the polymer around the filler particles and interfacial voids [97, 117, 124]. (5) Stability of MOFs in the used fluid (e.g., water or salt solution) is a steering tool that affects the stability and sustainability of the performance of the produced MMMs

Scaling up of MOFs synthesis to an industrial and commercial scale is becoming a more attractive trend recently. This is may be attributed to the very limited real life applications of MOFs-MMMs although there are some MOFs that are readily commercially available. The reason for this limited applications field is the use of hazardous reactants during MOFs synthesis and the difficult reaction conditions that limit their synthesis in pilot plants or at industrial scale [95].

Many approaches were suggested to address these previous challenges. On one hand, surveys by computational methods on the topological properties of the species should be carried out to select the most appropriate candidates for a certain application. Relation evaluation between molecular-level properties of MOFs and the MMMs performance and application is also another step on the road for selecting the most compatible MOFs for a certain polymer matrix. The polymer matrix choice itself is a point of equal importance [125]. On the other hand homogeneous MOFs dispersion in the polymer matrix can be achieved by reducing the filler particle size which will have more affinity with the polymeric matrix, hence improving the MOFs-MMMs performance. Decreasing the particle size will also increase the polymer/MOFs interface area, creating more selective pathways for species permeation. Surface modification is also another technique to improve the adhesion MOFs and the polymer. Surface functionalization can be carried out by introducing the appropriate functional group to the MOFs surface that is compatible with the polymer matrix [124]

Some of the future aspects that can be considered for further enhancement of the MOFs-MMM are: (1) Investigation of new synthesis techniques to manufacture MOFs-MMMs with oriented MOFs. Oriented MOFs will offer more facile pathways for the permeation of species from one side of the membrane to the other that is an important factor in processes like MD and pervaporation. (2) Incorporation of MOFs on the membrane surface without sacrificing their good adhesion to the polymer matrix in cases where the surface properties need to be more pronounced. (3) Conventional symthesis techniques lead to the perfect encapsulation of MOFs within the polymer matrix, hence decreasing the available surface area of MOFs for separation. Therefore, novel techniques introduction to bind the MOFs particles at a distance from the membrane surface may be the answer to increase the available MOFs surface area. (4) New synthesis routes at mild conditions and using non-hazardous materials should be investigated to solve the scaling up issue of the MOFs-MMMs”.

2.   It is necessary to provide a comprehensive table that compares the MOF-MMMs for aqueous applications. MOFs, polymer matrix, loadings, performance etc should be compared. 

Thank you so much for the valuable suggestion, a summary table (Table 2)  has been prepared and included, page 5

3. The different membrane preparation techniques for MOF-MMMs should be summarized and compared. A figure with several panels that show schematics of these techniques should be included in the manuscript.

Figures of different synthesis routes and  table to describe each process have been included in the manuscript.

4. The following MOF-MMM for water applications are missing from the review and should be discussed

( DOI: 10.1021/jacs.5b02276; 10.1039/C7TA00339K; 10.1021/acs.cgd.6b01398;  10.1021/acsanm.8b01902;10.1021/acsanm.8b00501; 10.1038/srep03740; 10.1016/j.memsci.2019.02.020.

All recommended references have been downloaded and read. Accordingly, the results have been discussed in the revised manuscript.

5.The manuscript lists most of the articles that fall within the scope but lacks critical assessment. The authors should add some critical comments as appropriate throughout the manuscript.

Several critical comments have been added to the revised manuscript

6. What are the main limitations of MOFs as fillers for MMMs? In depth discussions should be included in the manuscript. Stability issues and loss of performance as well as need for long-term performance studies should be mentioned.

Section 5 “MOFs-MMMs challenges, solutions and future prospective” has been added to the revised manuscript

7. Group large number of references should be avoided. These do not provide any useful information for the readers, e.g. [41-50].

References were regrouped in a proper way.

8. The issues of scale-up of both MOFs and MOF-based membranes need to be mentioned.

The issue was discussed in section 5

9. The figures taken from the literature are of low resolution. The authors should either redraw them or source a higher resolution image.

Better resolution images were added to the manuscript

10. The manuscript focuses on water application of MOFs. However, the authors should briefly mention in a sentence the potential and challenges of MOFs for non-aqueous MMMs and give some examples OSN

 (DOI: 10.1039/C4TA00628C;10.1039/C5RA10259F; 10.1016/j.memsci.2016.01.024)

 Potential and challenges are discussed in details. References are included in the manuscript.

11. The fabrication of green and sustainable membranes are becoming increasingly important. What is the potential of MOF-MMMs in that perspective? Are there any green MOFs? Have they been applied in any green process. These considerations should be added to the review.

Green MOFs was addressed:

A new approach in MOFs synthesis has emerged during the last decade which is the "green" synthesis of MOFs. The essential reason that triggered approach is that the conventional synthesis methods cannot be applied on large scale due to unsustainable and unsuitable synthesis conditions. In other words, the "green" approach emerged essentially from the need of cheap, renewable, avoiding waste while saving energy and recyclable starting materials to be applied industrially. The term "green" in MOFs synthesis can be described in the following simple points: (1) Avoid excessive use of solvents during synthesis or fabrication so as to minimize waste considerably. (2) Highly selective and high through output synthesis techniques should be adopted to maximize production and avoid any by-products. (3) Nontoxic and harmless reactants to the human health and environment and simple precursors should be used in synthesis techniques whenever applicable and practical. (4) The synthesis routes should preserve the targeted properties of the products. (5) Synthesis techniques employing low energy requirements should be applied where MOFs would be fabricated at atmospheric pressure and ambient temperature. (6) Renewable sources for the organic linkers (such as starch and cellulose) and solvents should be used whenever technically and economically viable. (7) Products should not be environmentally persistent and to be degraded under working conditions. (8) Development of new controllable synthesis techniques comprising shorter reaction times to avoid formation of side products as much as possible. (9) During scaling up MOFs synthesis, development of new chemical methods should be chosen so as to avoid chemical accidents such as explosions and releases [95].

Green MOFs has been applied as a green template for the formation of macropores to produce porous matrix membranes (PMM) with enhanced porosity for water treatment [101,106] MOFs-MMM syntheses routes can be categorized into four general routes; blending figure 1, in situ growth figure 3, Layer by layer (LBL) figure 5 and gelatin-assisted seed growth figure 6. Table 1 summarizes the procedure of these four routes”.

12. The conclusion section is vague. More tangible information that guide the readers on structure-function-performance relationship should be provided. The most promising MOFs for aqueous applications should also be highlighted here.

The conclusion section has been rewritten to give a more comprehensive discussion.

Since the manufacture of the first membrane, many efforts were spent on enhancing the membrane properties whether thermal, mechanical or fouling resistance. MOFs are a class of hybrid materials made up of organic ligands and metal clusters that possess huge surface areas and exceptional properties. MOFs-based MMMs were introduced to solve different membrane issues as well as the trade-off issue between selectivity and permeability of polymeric membranes. MOFs organic ligands increased MOFs compatibility with the polymeric matrices, hence well dispersion of the MOFs fillers is achieved. However, increasing MOFs loading beyond 10% in most cases led to formation of aggregates which had a significant effect on the membrane performance. Aggregates increase the membrane pore sizes, consequently form nonselective voids which increased water and salt permeation, hence decreased the salt rejection. MOFs structure and properties and the way the filler is incorporated in the polymeric matrix also affected the MOFs-MMMs properties and performance. On one hand, membrane hydrophobicity/hydrophilicity is dependent on the MOFs itself whether it is hydrophobic or hydrophilic in nature and if it was perfectly encapsulated by the membrane matrix or deposited/migrated to the surface. Water preferred to be transported through hydrophilic MOFs such as MILs and UiO-66. On the contrary water was transported through preferential path ways around hydrophobic MOFs such as ZIFs. On the other hand, geometry (window size) affected the water permeation and salt passage, where species with smaller molecular size than the MOFs window size can pass through the MOFs structure and those with bigger molecular size are rejected. Different synthesis techniques were investigated for the manufacture of MOFs- based MMMs. For example In situ growth technique produced MOFs-MMMs with better dispersion and compatibility, while gelatin-assisted seed growth produced a uniform crack free thin film MOFs layer at room temperature. Despite all the merits of these hybrid membranes, it has its own complexities and difficulties that restricted its large scale application and fabrication. Some of which are the high cost, difficult and dangerous MOFs synthesis techniques and  stability and sustainability of performance. Additionally the use of MOFs in membranes used in water purification applications may be considered as a potential environmental and health hazard. Therefore new synthesis techniques with mild conditions and utilizing nonhazardous compounds, selection of a compatible MOF/polymer system and the specific properties that need to be promoted for a definite application should be well investigated. To conclude MOFs-MMMs have a great potential in different separation application due the exceptional properties offered by MOFs but its success, competitiveness and upscaling needs further persistent efforts to solve problems identified with their fabrication and application”.

Reviewer 2 Report

This work provides a summary of the recent use of MOFs as fillers to modify membrane performance in water purification, mainly in the pressure-driven process such as reverse osmosis. The title suggests that the authors are focusing only on mixed matrix systems and process such as RO, NF, UF and not pervaporation or distillation. This is ok as long as there are sufficient pieces of literature to justify a review. To improve the review, the followings are suggested:

1) It appears that some key studies are not included in this work; for example:

Filler: ZIF-L

Low, Z.X., Razmjou, A., Wang, K., Gray, S., Duke, M. and Wang, H., 2014. Effect of addition of two-dimensional ZIF-L nanoflakes on the properties of polyethersulfone ultrafiltration membrane. Journal of membrane science460, pp.9-17.

Fillers: ZIF-8, MIL-53 (Al), NH2-MIL-53 (Al) and MIL-101 (Cr). Although aimed at OSN, it is still a polyamide system.

Sorribas, S., Gorgojo, P., Téllez, C., Coronas, J. and Livingston, A.G., 2013. High flux thin film nanocomposite membranes based on metal–organic frameworks for organic solvent nanofiltration. Journal of the American Chemical Society135(40), pp.15201-15208.

2) Page 3 Line 126- Ref [32] should be Ref [33]. Please confirm

3) Page 5 Line 200- This is not mixed matrix system

4) It would be useful to include a summary table or figure to summarize all work reviewed in this work as there are not many. To make a fair comparison, the filler loading should be included. Membrane performance should be reported in the same unit, e.g. LMH/bar.

5) It would also be useful for readers if the selection criteria are included and a perspective/future direction from the authors are provided. E.g., MOF stability in water or in salt solution, size of MOF, processibility, scalability etc.

The above would make this review more useful to the readers and compliment reviews that were written in this area.

Author Response

Dear Dr. Angela Yang,

Thank you very much for the constructive and interesting reviewers’ comments on submitted manuscript that will certainly help to enhance the quality of the manuscript. I am pleased to inform you that the revised manuscript has been updated according to the reviewers’ comments and addressed each comment separately. The reviewers’ comments and answers are listed below.

Reviewer#2

Comments and Suggestions for Authors

This work provides a summary of the recent use of MOFs as fillers to modify membrane performance in water purification, mainly in the pressure-driven process such as reverse osmosis. The title suggests that the authors are focusing only on mixed matrix systems and process such as RO, NF, UF and not pervaporation or distillation. This is ok as long as there are sufficient pieces of literature to justify a review. To improve the review, the followings are suggested:

Thank you for your valuable comments which helped in improving the manuscript. Regarding pervaporation and distillation processes, they are mentioned in references  [104,38]

1) It appears that some key studies are not included in this work; for example:

Filler: ZIF-L

Low, Z.X., Razmjou, A., Wang, K., Gray, S., Duke, M. and Wang, H., 2014. Effect of addition of two-dimensional ZIF-L nanoflakes on the properties of polyethersulfone ultrafiltration membrane. Journal of membrane science, 460, pp.9-17.

Fillers: ZIF-8, MIL-53 (Al), NH2-MIL-53 (Al) and MIL-101 (Cr). Although aimed at OSN, it is still a polyamide system.

Sorribas, S., Gorgojo, P., Téllez, C., Coronas, J. and Livingston, A.G., 2013. High flux thin film nanocomposite membranes based on metal–organic frameworks for organic solvent nanofiltration. Journal of the American Chemical Society, 135(40), pp.15201-15208.

References are discussed and included in the manuscript [84 & 109]

2) Page 3 Line 126- Ref [32] should be Ref [33]. Please confirm

All the references have been checked and corrected

3) Page 5 Line 200- This is not mixed matrix system

In full agreement with your opinion, this reference was replaced by a more appropriate one

4) It would be useful to include a summary table or figure to summarize all work reviewed in this work as there are not many. To make a fair comparison, the filler loading should be included. Membrane performance should be reported in the same unit, e.g. LMH/bar.

Thank you so much for the valuable suggestion, a summary table has been prepared and included at the end of the manuscript, page 5 – 14.

5) It would also be useful for readers if the selection criteria are included and a perspective/future direction from the authors are provided. E.g., MOF stability in water or in salt solution, size of MOF, processibility, scalability etc.

Similar to our reply on the first comment by reviewer 1, a new section has been added “section 5”.

The above would make this review more useful to the readers and compliment reviews that were written in this area.

Round  2

Reviewer 1 Report

The authors have done a thorough revision of the manuscript, which is now ready to be published. 

Author Response

Manuscript ID: Membranes-507346

Dear Dr. Angela Yang,

Thank you very much for the constructive and interesting reviewers’ comments on submitted  manuscript. We have revised the manuscript in consideration of the comments of the reviewer. The comments of reviewers are encouraging and they have been very valuable in improving the quality of the manuscript. The changes and suggested corrections have been made in blue color and the list of changes made in relation to each point together with our answers/clarifications are presented in subsequent section (Response to Comments).  We hope that the revised manuscript will be accepted for publication. We look forward to your positive response.

Response to comments

1-1-  The authors have replied to most of the reviewers comments and should be taken into account. I am not sure they have provided a critical personal assessment as suggested by the reviewers though. Besides, it is weird such emphasis of the MOF-based membranes in pervaporation when their intended aim is focusing in water purification, and reviews of MOF mixed matrix membranes are not cited, such as:
(i)https://pubs.rsc.org/en/content/articlelanding/2010/EE/b923980b#!divAbstract
(ii)https://www.researchgate.net/publication/259770693_Fabrication_of_Porous_Matrix_Membrane_PMM_Using_Metal 

Reply #1:

-The reference has been discussed and cited in the manuscript as reference number 96:

(i)  https://www.researchgate.net/publication/259770693_Fabrication_of_Porous_Matrix_Membrane_PMM_Using_Metal_Organic_Framework_as_Green_Template_for_Water_Treatment/citation/download.

-Computational methods on the topological properties of membrane was pointed out using the abovementioned reference and cited as reference number 127:

(ii)   https://pubs.rsc.org/en/content/articlelanding/2010/EE/b923980b#!divAbstract 

2- Figure 7 does not have any citation reference, does it belong to the authors' own work?

Reply #2

Thank you very much for reviewer’s observation. Citation reference of figure 7 has been added

3- The issue of the "green" MOFs and the water resistant MOFs should be further clarified. Perhaps this is related to the lack of scalability of the MOF-based MMMs in particular, while the lack of scalability of MMMs is also patent because of the lack of adhesion between filler and components, as the authors are well aware of. Please rephrase.

Reply #3

Further clarifications have been added to the revised manuscript in different sections:

Section2:

For example, green techniques were used to synthesis green ZIF-8. Alternative less hazardous solvents such as methanol was used instead of DMF. A major advantage of this approach is inly water was formed as byproduct. However, the porosity of the formed ZIF-8 was much lower than expected. Mechanical techniques by using ball milling were also reported for the synthesis of ZIF-8. The issue here was that reaction took very long time to produce ZIF-8 with the highest porosity and that a core of zinc oxide was found within the ZIF-8 shells, hence the reaction was not fully achieved. However, all the reported procedures utilizing lower amounts of solvents or no solvent at all requires the removal of residual linkers by solvent treatment after synthesis [95

Section 5:

5) The difficulty of scalability of the MOFs-based MMMs due to the insufficient adhesion between the MOFs as fillers and the polymer matrix

4- Careful revision of the English expression is highly recommended. There are many mistakes especially in the red-recently added part that looks as precipitation.

Reply #4

Manuscript has been revised carefully. Typos and other minor corrections have been made

Reviewer 2 Report

The authors made a significant revision to the previous draft. Happy for the manuscript to be published as is.

Author Response

Manuscript ID: Membranes-507346

Dear Dr. Angela Yang,

Thank you very much for the constructive and interesting reviewers’ comments on submitted  manuscript. We have revised the manuscript in consideration of the comments of the reviewer. The comments of reviewers are encouraging and they have been very valuable in improving the quality of the manuscript. The changes and suggested corrections have been made in blue color and the list of changes made in relation to each point together with our answers/clarifications are presented in subsequent section (Response to Comments).  We hope that the revised manuscript will be accepted for publication. We look forward to your positive response.

Response to comments

1-1-  The authors have replied to most of the reviewers comments and should be taken into account. I am not sure they have provided a critical personal assessment as suggested by the reviewers though. Besides, it is weird such emphasis of the MOF-based membranes in pervaporation when their intended aim is focusing in water purification, and reviews of MOF mixed matrix membranes are not cited, such as:
(i)https://pubs.rsc.org/en/content/articlelanding/2010/EE/b923980b#!divAbstract
(ii)https://www.researchgate.net/publication/259770693_Fabrication_of_Porous_Matrix_Membrane_PMM_Using_Metal 

Reply #1:

-The reference has been discussed and cited in the manuscript as reference number 96:

(i)  https://www.researchgate.net/publication/259770693_Fabrication_of_Porous_Matrix_Membrane_PMM_Using_Metal_Organic_Framework_as_Green_Template_for_Water_Treatment/citation/download.

-Computational methods on the topological properties of membrane was pointed out using the abovementioned reference and cited as reference number 127:

(ii)   https://pubs.rsc.org/en/content/articlelanding/2010/EE/b923980b#!divAbstract 

2- Figure 7 does not have any citation reference, does it belong to the authors' own work?

Reply #2

Thank you very much for reviewer’s observation. Citation reference of figure 7 has been added

3- The issue of the "green" MOFs and the water resistant MOFs should be further clarified. Perhaps this is related to the lack of scalability of the MOF-based MMMs in particular, while the lack of scalability of MMMs is also patent because of the lack of adhesion between filler and components, as the authors are well aware of. Please rephrase.

Reply #3

Further clarifications have been added to the revised manuscript in different sections:

Section2:

For example, green techniques were used to synthesis green ZIF-8. Alternative less hazardous solvents such as methanol was used instead of DMF. A major advantage of this approach is inly water was formed as byproduct. However, the porosity of the formed ZIF-8 was much lower than expected. Mechanical techniques by using ball milling were also reported for the synthesis of ZIF-8. The issue here was that reaction took very long time to produce ZIF-8 with the highest porosity and that a core of zinc oxide was found within the ZIF-8 shells, hence the reaction was not fully achieved. However, all the reported procedures utilizing lower amounts of solvents or no solvent at all requires the removal of residual linkers by solvent treatment after synthesis [95

Section 5:

5) The difficulty of scalability of the MOFs-based MMMs due to the insufficient adhesion between the MOFs as fillers and the polymer matrix

4- Careful revision of the English expression is highly recommended. There are many mistakes especially in the red-recently added part that looks as precipitation.

Reply #4

The manuscript has been revised carefully. Typos and other minor corrections have been made
